# Molecular architecture and functional dynamics of the pre-incision complex in nucleotide excision repair

Jina Yu [1,2], Chunli Yan[1,2], Tanmoy Paul[1,2], Lucas Brewer [1,2], Susan E. Tsutakawa [3], Chi-Lin Tsai [4], Samir M. Hamdan [5], John A. Tainer [3,4] ✉ & Ivaylo Ivanov [1,2] ✉

Nucleotide excision repair (NER) is vital for genome integrity. Yet, our understanding of the complex NER protein machinery remains incomplete. Combining cryo-EM and XL-MS data with AlphaFold2 predictions, we build an integrative model of the NER pre-incision complex(PInC). Here TFIIH serves as a molecular ruler, defining the DNA bubble size and precisely positioning the XPG and XPF nucleases for incision. Using simulations and graph theoretical analyses, we unveil PInC's assembly, global motions, and partitioning into dynamic communities. Remarkably, XPG caps XPD's DNA-binding groove and bridges both junctions of the DNA bubble, suggesting a novel coordination mechanism of PInC's dual incision. XPA rigging interlaces XPF/ERCC1 with RPA, XPD, XPB, and 5' ssDNA, exposing XPA's crucial role in licensing the XPF/ERCC1 incision. Mapping disease mutations onto our models reveals clustering into distinct mechanistic classes, elucidating xeroderma pigmentosum and Cockayne syndrome disease etiology.

Nucleotide excision repair (NER) is a genome maintenance pathway critical for human health. NER repairs a vast array of structurally unrelated DNA lesions caused by ultraviolet radiation, reactive oxygen species, environmental carcinogens, and chemotherapeutic agents such as cis-platinum[1-4]. The variety of lesions processed by NER has led to the evolution of a complex protein machinery[5] that performs damage recognition, DNA unwinding, damage verification, precise dual incision of lesioned DNA, and gap-filling DNA synthesis. Despite numerous biochemical and genetic studies, knowledge of the inner workings of this machinery remains fragmentary. Predictably, functional impairment of NER complexes by mutations causes severe human genetic diseases[6-11]—xeroderma pigmentosum (XP) and Cockayne syndrome (CS). The vastly divergent disease phenotypes range from normal development with an extreme cancer predisposition to developmental abnormalities, accelerated neurodegeneration, and premature ageing without cancer predisposition. Yet, a molecular basis for this striking heterogeneity in clinical outcomes has not emerged.

There are two NER sub-pathways—global genome NER (GG-NER) and transcription-coupled NER (TC-NER). GG-NER is transcription-independent and broadly probes the genome for damage. By contrast, TC-NER is specialized in bulky lesion removal from actively transcribed DNA[12-14]. The two sub-pathways differ in the early damage recognition steps but converge prior to lesion scanning and incision. Therefore, here we focus exclusively on the GG-NER sub-pathway. In GG-NER, the lesion recognition factor XPC aided by human Rad23B and Centrin 2 (CETN2)[15-17] scans the genome for DNA damage and in a twist-to-open mechanism exposes the lesion, creating a nascent DNA bubble[18-21]. Afterward, transcription factor IIH (TFIIH)[11,22-29] is recruited to the damaged site, forming a pre-unwound NER complex[15,16]. TFIIH contains

[1]Department of Chemistry, Georgia State University, Atlanta, GA, USA. [2]Center for Diagnostics and Therapeutics, Georgia State University, Atlanta, GA, USA. [3]Molecular Biophysics and Integrated Bioimaging, Lawrence Berkeley National Laboratory, Berkeley, CA, USA. [4]Department of Molecular and Cellular Oncology, The University of Texas MD Anderson Cancer Center, Houston, TX, USA. [5]Bioscience Program, Biological and Environmental Sciences and Engineering Division, King Abdullah University of Science and Technology (KAUST), Thuwal, Saudi Arabia. ✉e-mail: jtainer@mdanderson.org; iivanov@gsu.edu

ten protein subunits – seven forming core TFIIH (XPD, XPB, p44, p34, p8, p62, and p52) and three comprising the CAK complex (MAT1, Cdk7 and Cyclin H)[30]. While CAK is key for TFIIH's function in transcription, its dissociation from core TFIIH is required for functional NER. Two subunits, XPB and XPD, are DNA translocases[31] with key functional roles in DNA remodeling during NER. Assisted by XPA[32–34], XPB first unwinds the DNA duplex upstream of the lesion, expanding the NER bubble[35]. XPD then verifies the lesion[31,34,36] by scanning the damage-containing DNA strand. Since NER accuracy is paramount for genome integrity, all these dynamic reversible transformations must be completed before actual lesion removal[37]. DNA damage removal relies on the coordinated action of two structure-specific endonucleases, XPG and XPF/ERCC1[38–44]. XPG is a member of the FEN1 (flap endonuclease 1) family of nucleases[38]. The enzyme recognizes the 3′ junction of the NER bubble and cleaves the damaged strand one nucleotide into the DNA duplex. Correspondingly, XPF belongs to the Mus81 nuclease family and functions as a stable heterodimer with the ERCC1 protein[43]. The XPF/ERCC1 complex acts on the 5′ junction, making an incision 3–5 nucleotides into the DNA duplex. Strand incision constitutes a point of no return[4] and commits the NER machinery to complete the repair process. Failure to conclude repair post-incision has dire consequences for the cell as the resulting abortive intermediates are more mutagenic or lethal than the original DNA damage. To prevent unlicensed incisions, NER relies on an intricately orchestrated assembly of a pre-incision complex (PInC)[4,5,42] comprised of TFIIH, XPG, XPF/ERCC1, XPA, and RPA. Within PInC, the coordinated action of XPG and XPF/ERCC1 on both sides of the lesion results in a ~ 27-nucleotide ssDNA gap substrate[45]. Subsequently, Polδ, RFC, and PCNA are loaded at the 5′ junction of this substrate to initiate gap-filling DNA synthesis and restore the excised region. Lastly, DNA ligase seals the nicked DNA to complete the repair process. Correct assembly of the PInC is critical for the accuracy of the dual incision process and, therefore, vital to avoid genomic injury.

Here we have used integrative molecular modeling to synthesize available structural data on NER constituent proteins and subassemblies and combined it with biochemistry and cross-linking mass spectrometry (XL-MS) restraints to create a practically complete structural model of the human NER pre-incision complex. The new model enables comprehensive side-by-side structural comparison to the previously determined lesion scanning complex (LSC), thus shedding light on the reorganization of the NER protein machinery from the middle through the late stages of the pathway. The model also serves as a starting point for microsecond-timescale molecular dynamics (MD) simulations, which reveal PInC's global motions. We also employ graph-theoretical algorithms to partition the assembly into dynamic communities and network analysis to reveal the key differences between the functional dynamics of the pre-incision and lesion-scanning complexes. Importantly, mapping of disease mutations onto our network models allows us to cluster the mutants into distinct mechanistic classes impacting DNA binding, protein stability, and dynamics at PInC protein interfaces. Thus, our integrative modeling provides a roadmap for future experiments to test the interplay between the structural disruption of NER complexes by mutations and the etiology of devastating genetic diseases. Collectively, our results unveil how the NER machinery dynamically reshapes itself and self-regulates to achieve precise lesion removal and preserve genome integrity.

## Results

### TFIIH serves as a molecular ruler defining the NER bubble size and PInC's overall organization

We synthesize diverse structural data to create an integrative model of the most crucial intermediate in NER—the pre-incision complex. To construct the model, we systematically evaluated the cryo-EM structures and densities of human apo-TFIIH[22], TFIIH/XPA/DNA[46], XPF/ERCC1[43], and XPA/ERCC1[47] along with the X-ray structures of the XPG catalytic core[38]

and RPA–ssDNA (RPA70, RPA32 and RPA14)[48]. Missing unmodelled regions that could be traced in the respective EM densities were built de novo. Predicted folded protein regions not resolved in the experimental structures were modeled with AlphaFold[49,50]. Inclusion of these structured regions was necessary for the molecular dynamics simulations to unravel the functional dynamics of the pre-incision complex. Positioning of the newly modeled modules and protein domains within the PInC assembly was guided by existing mutational and cross-linking mass spectrometry (XL-MS) data[46] on NER protein interfaces and by AlphaFold2-multimer predictions. Our combined analysis yields a defined structural model of the pre-incision complex (Fig. 1 and Supplementary Movie 1) and provides unanticipated insights into functional interactions of its constituent proteins.

The pre-incision complex is organized around transcription factor IIH (TFIIH)[11,22–27,30], which serves as a platform for the assembly and reorganization of the NER machinery. TFIIH's adaptable modular architecture underpins its multiple cellular functions in GG-NER, TC-NER, and transcription initiation. While the structure of the pre-unwound NER complex (TFIIH/XPC/HRAD23/CETN2/DNA)[15,16] closely resembles the open subunit arrangement of apo-TFIIH[22], the PInC features a closed circular arrangement of the seven core TFIIH subunits (Fig. 1a). The two translocase subunits, XPB and XPD, are bound to a DNA bubble substrate and share an extended interface braced by contacts with the p44 subunit. This interface is essential for the sequential coordination of the XPB and XPD activities during the transitions of the NER protein machinery from DNA unwinding to lesion scanning and then to strand incision[37]. The closed circular TFIIH architecture imparts structural rigidity, making TFIIH into a stable platform for the assembly of XPF/ERCC1, XPG, XPA, and RPA (Fig. 1a).

TFIIH also serves as a molecular ruler defining the spatial extent of the NER bubble (Fig. 1a and c) and the relative positioning of the XPG and XPF nucleases (Fig. 1a). The DNA bubble region of PInC has 23 nucleotides between the 3′ and 5′ junctions, matching the most probable length (27 nucleotides) of the excision products (Supplementary Note 1). Thus, our model explains the remarkable precision of the NER dual incision[51] as observed in NER gel assays.

In turn, the small size of the DNA bubble dictates the exceedingly compact spatial arrangement of XPF/ERCC1, XPG, XPA, and RPA in the PInC. Notably, the XPF/ERCC1 complex is positioned on the anterior side of TFIIH's XPD subunit, capping the 5′ DNA junction just above the XPB–XPA interface. Correspondingly, XPB binds the duplex leading into the 5′ junction and accommodates dsDNA in a groove between its RecA-like ATPase domains. In this orientation, the XPF nuclease active site is poised for incision into the 5′ duplex. Conversely, our PInC model places the XPG catalytic core at the extreme opposite end of the NER bubble. The XPG core binds the 3′ junction, facing the posterior side of XPD. RPA inserts itself between the XPG core and the XPA DNA-binding domain and protects the undamaged DNA strand. Remarkably, the spatial extent of XPG between its catalytic core and the XPG anchor domain (residues 157–296) is such that the nuclease spans the entire distance between the 3′ and the 5′ DNA junctions (Fig. 1a and b). Thus, XPG stretches along the XPD–RPA interface and caps XPD's DNA-binding groove. A critical component of the PInC, XPA extends from the RPA trimer core to TFIIH's p8 subunit, making extensive contacts with DNA, XPD, XPB, and XPF/ERCC1. XPA also provides a β-hairpin, which separates the two DNA strands at the 5′ junction[46]. The damaged strand passes through XPD while the non-damaged strand is directed toward the XPA zinc-finger domain and RPA. Notably, our integrative model shows that XPG and XPF/ERCC1 can both be accommodated in catalytically competent orientations, providing a structural basis for the coordinated dual incision activity of PInC.

### XPG is positioned for incision at the 3′ DNA junction of PInC

We first modeled the XPG nuclease catalytic core[41] on the back side of XPD poised for incision at the 3′ ss/dsDNA junction. XPG is a member

of the 5′ nuclease family that includes FEN1, EXO1, and GEN1. A defining feature in these enzymes is a two-helix gateway or Arch domain, by which the nuclease selects for a single-stranded DNA feature in its substrate. In FEN1, it "threads" DNA flaps with free ends through a "needle" hole formed by this gateway and a helical cap[52]. Like FEN1, XPG harbors a helical arch (Fig. 2a and b) comprised of two gateway helices, GH1 (residues 33–41) and GH2 (residues 82–129). GH2 also forms an extended coiled coil with a capping helix CH (residues 734–763) as predicted by AlphaFold2 calculations. Unlike FEN1, XPG acts on a bubble substrate. This imposes a topological restriction on the gateway helices interacting with DNA, making it impossible to 'thread'. Thus, it is key that we could model the single-stranded DNA slotting between the AlphaFold2-predicted XPG gateway helices (Fig. 2a and b). In this positioning, the coiled-coil region clamps down on top of the lesion-containing strand.

## XPG is poised to facilitate DNA strand separation during XPD unwinding

XPG's conserved structural elements (Fig. 2b and Supplementary Fig. 1) are ideally positioned not just for incision[41] but also for strand separation. A hallmark of 5′ nucleases is that DNA binding is dominated by duplex DNA interaction. Thus, in our model, XPG engages the 3′ duplex of the NER bubble substrate via a conserved electrostatically compatible surface suitable to recognize the helical features of dsDNA (Supplementary Fig. 2). The XPG active site is situated in a groove near the center of enzyme (Fig. 2b). On one side of the active site, ssDNA is perched on a K+ ion coordinated by the helix-2-turn-helix (H2TH) motif (residues 848–880). DNA binding is reinforced through XPG-specific contacts to an α12b motif (residues 912–918) through a basic patch (K913, K916, and K917). On the other side of the active site, a hydrophobic wedge motif (residues 31–67) and a β-hairpin motif (residues 820–836) form a positively-charged groove, expanding the DNA-binding surface (Fig. 2b and Supplementary Fig. 2). Relevant to both XPG nuclease activity and XPD translocase activity, the

hydrophobic wedge and β-hairpin motifs ensure placement of the 3′ junction with a ~100° DNA bending angle. Based on work with FEN1, this substrate is validated by having single-stranded DNA between the gateway helices and junction DNA at the hydrophobic wedge prior to incision[41,53]. Importantly, in this position the XPG hydrophobic wedge and β-hairpin motifs could act as a helicase pin to facilitate strand separation during XPD's DNA unwinding activity.

In our PInC model, we maintained the gateway arch angle with the XPG core. In this arrangement, the core makes minimal contacts with TFIIH. We also present an alternative model (Supplementary Fig. 3) where a modest adjustment to the gateway arch orientation allows the XPG core to tilt and make contacts with the XPD Arch domain. As the TFIIH/XPA/DNA structure was obtained without a true bubble substrate, no such contacts are present in the XL-MS data. Yet, the possibility cannot be excluded as XPD was shown to stimulate XPG incision after completion of lesion scanning[54].

Bulky lesion blockage within XPD serves as the basis for lesion verification in NER. XPG cleaves DNA with modest variability, typically 5–8 nucleotides from a bulky lesion[51,55]. In our models, such blockage occurs 8 nucleotides from the 3′ junction at a constriction in the XPD's DNA-binding groove proximal to His135 (Fig. 2f) and is accompanied by closure of the Fe-S and Arch domain spacing around the lesion. Thus, we posit a mechanism where termination of XPD's unwinding activity enables XPG repositioning for productive 3′ incision.

## XPG bridges the two DNA junctions offering an unexpected 5′ incision sensing mechanism

The XPG nuclease has a highly unusual architecture[41]—its catalytic core is formed by an N-terminal (N) and internal (I) nuclease domains, which are proximal in structure but distal in sequence, separated by ~600-residue insertion denoted as the "spacer region" or R-domain (Fig. 2b). The I- and N-domains exhibit high conservation to flap endonuclease 1[52] and other members of the FEN1 nuclease family (Supplementary Fig. 1). By contrast, the spacer region is not conserved with other FEN1

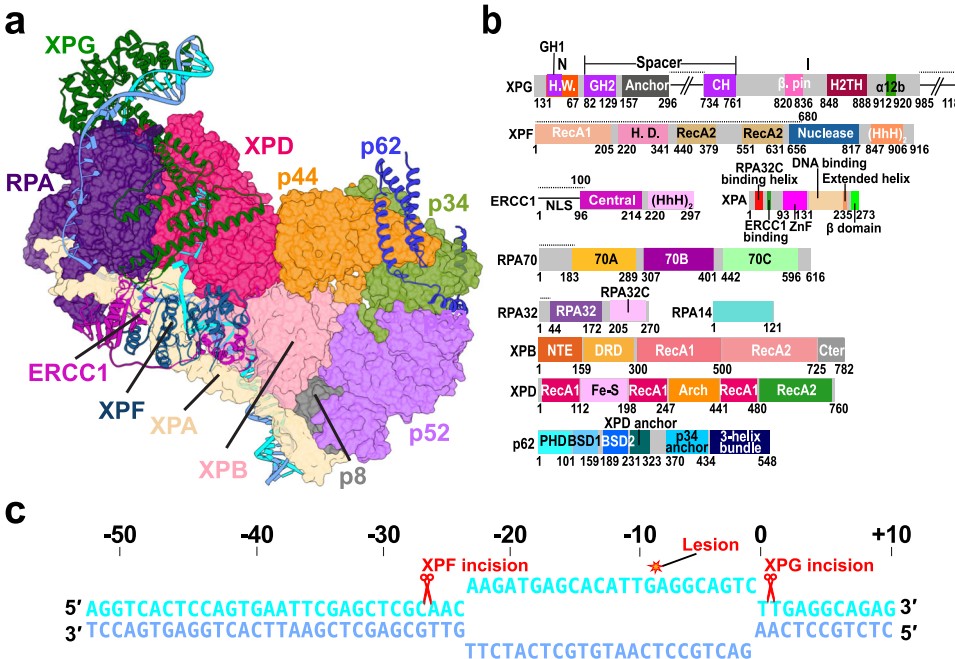

**Fig. 1 | Integrative model of the PInC unveils the overall structural organization of the assembly. a** View of the PInC assembly colored by subunits. XPG, XPF/ERCC1, p62, and DNA are shown in cartoon representation. TFIIH, XPA, and RPA are shown in surface representation. The lesion containing DNA strand is shown in cyan; the undamaged strand is shown in blue. **b** Domain organization of the PInC constituent proteins XPG, XPF, ERCC1, XPA, RPA, XPB, XPD, and p62 mapped onto

their respective sequences. Abbreviations denote H.W.—hydrophobic wedge; GH—gateway helix; CH—coiled-coil helix; H2TH—helix-2-turn-helix; H.D.—helicase domain; HhH—helix-hairpin-helix; DRD—damage recognition domain; NTE—N-terminal extension. **c** Schematic showing the DNA substrate of PInC, the length of the NER bubble, and the positions of the lesion site (red star) and the two incision sites (red scissor symbols).

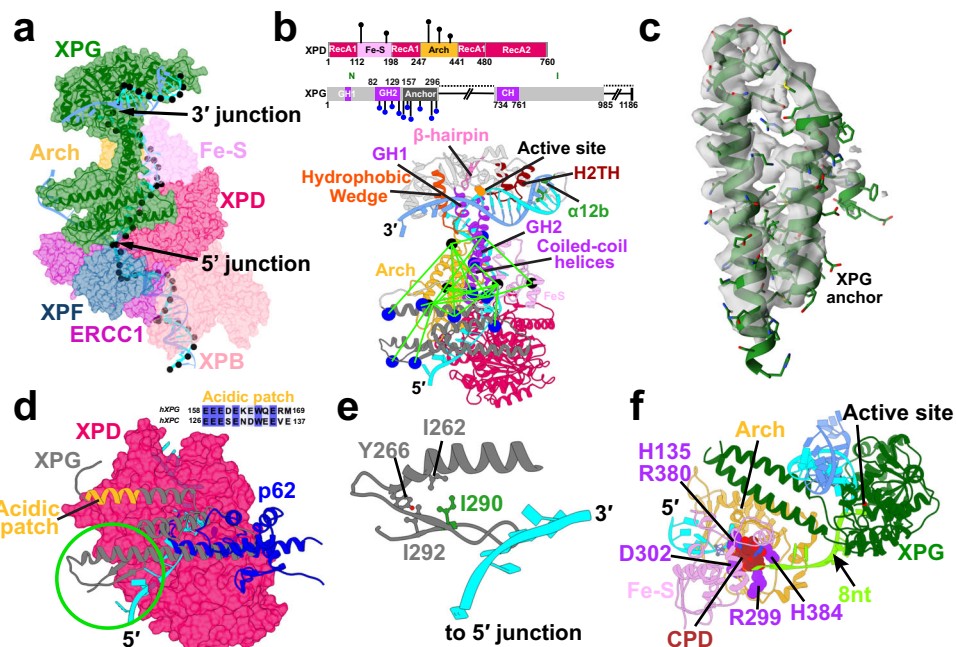

**Fig. 2 | The XPG nuclease bridges the opposing ends of the NER bubble, providing an unexpected 5′ incision sensing mechanism. a** View of XPG at the 3′ DNA junction. XPG is depicted as cartoon and transparent surface and colored in dark green. The path of the DNA through TFIIH's XPD subunit is shown by black spheres placed at the phosphorus atoms. **b** Crosslinks between XPG and XPD mapped onto the PInC structure. The XPD Arch and Fe-S domains are colored in orange and purple, respectively. XPG and XPD residues participating in the crosslinks are depicted as blue and black spheres, respectively. A schematic of the crosslinks mapped onto XPD's and XPG's sequences is shown above. Conserved structural motifs in the XPG catalytic core are shown; the two XPG coiled-coil helices are shown in purple; the XPG anchor domain is shown in gray. **c** The XPG anchor domain fitted into a segment from the TFIIH/XPA/DNA cryo-EM density (EMDB accession code: EMD-4970). **d** Overlay of AlphaFold2 docking models of the XPG-anchor and the p62 XPD-anchor with XPD. The β-hairpin, packed against the XPG-anchor domain is highlighted by a green circle. The acidic patch of XPG is highlighted in orange. The inset displays the sequence alignment of the acidic patches of human XPG and XPC. **e** Zoomed-in view of the β-hairpin interacting with the XPD anchor and ssDNA at the 5′ junction. A hydrophobic cluster involving residue I290 colored in green. I290N substitution is a recognized XP/CS disease mutant. **f** A CPD lesion blocked inside XPD's DNA-binding groove near His135 and 8 nucleotides away from the 3′ junction.

family enzymes, suggesting it evolved to fulfill functional requirements specific to XPG. The uniquely extended spacer region has been shown biochemically as important in establishing XPG's specificity for ssDNA−dsDNA junctions[56]. Additionally, structured elements of the spacer mediate protein interactions essential for NER progression: e.g., XPG contacts with TFIIH's XPD, XPB, p62, and p44 subunits as well as contacts with the RPA DNA-binding domains[46,57–60]. No experimental structures are available for any part of the spacer region. Therefore, we used AlphaFold2 to determine predicted folded segments within the XPG spacer region (Fig. 2). As there is no substrate-bound XPG structure, the XPG-bound DNA was based on the FEN1-bound substrate structure (PDB ID:5UM9)[61].

The TFIIH/XPA/DNA cryo-EM structure[46] was pivotal for our hybrid modeling as it offered an unprecedented molecular view of the interactions of the core TFIIH subunits during lesion scanning. While XPG was not modeled in the electron density, the study provided valuable cross-linking data[46]. The XL-MS data suggests XPG remains flexibly tethered on the back side of XPD close to the ssDNA exit point between the XPD Arch and Fe-S domains. By carefully examining all XPG-TFIIH crosslinks and using AlphaFold2[49,50] to guide the modeling protocol, we were able to position all structured segments of XPG relative to TFIIH. Thus, placing the XPG coiled-coil helices between the Fe-S and Arch domains of XPD (Fig. 2b) is consistent with the cross-linking data[46]. AlphaFold2 also identified a previously unrecognized helical bundle domain of XPG (residues 157–296), which serves to anchor XPG to XPD and displace the p62 XPD-anchor and BSD2 domains. Remarkably, we were able to perfectly fit this domain into the cryo-EM density of the TFIIH/XPA/DNA complex (Fig. 2c), matching even the side chain densities. In Fig. 2d we show an overlay of

XPG−XPD and p62−XPD models generated with AlphaFold2-multimer. The overlay shows that the XPG-anchor domain and the p62 XPD-anchor compete for the same binding site on the XPD surface, consistent with a recent study which modeled p62-XPD interactions.[62] The cryo-EM density unambiguously shows that the XPG-anchor domain is bound to XPD. Yet, the XL-MS data, in addition to 23 above-threshold XPD−XPG crosslinks, also features 4 p62−XPD crosslinks. This may indicate that 1) XPG and p62 compete for the same XPD binding site, or 2) p62's anchor helices and BSD2 domain remain flexible near XPD's Fe-S domain after displacement. Since XL-MS is subject to sample conformational and compositional variability, presence of minor p62-bound species different from the dominant cryo-EM structure cannot be excluded−a point that remains to be addressed by future studies. Notably, P62 displacement from TFIIH is only partial−p62's p34-anchor and 3-helix bundle domains (Fig. 1a) are clearly resolved in the TFIIH/XPA/DNA EM density and their placement is fully supported by the XL-MS data.

With the XPG-core and XPG-anchor domains bridged by the two coiled-coil helices, XPG forms a bi-lobed structure whose spatial extent perfectly matches the distance between the 3′ and the 5′ DNA junctions. XPG has been shown to bind early in the PInC assembly process and stabilize the NER bubble. Yet, the first incision occurs 5′ to the lesion by the action of XPF/ERCC1. In this "XPG binds first, cleaves last" model, the two nucleases are expected to sequentially coordinate their activities. However, no molecular mechanism for such coordination had emerged. Remarkably, our PInC model suggests that XPG serves as a bridge between the opposite ends of the NER bubble, reaching all the way to the XPF/ERCC1 interface (Fig. 2a and b). The XPG-anchor domain also inserts a β-hairpin (residues 262–296) near the 5′ junction,

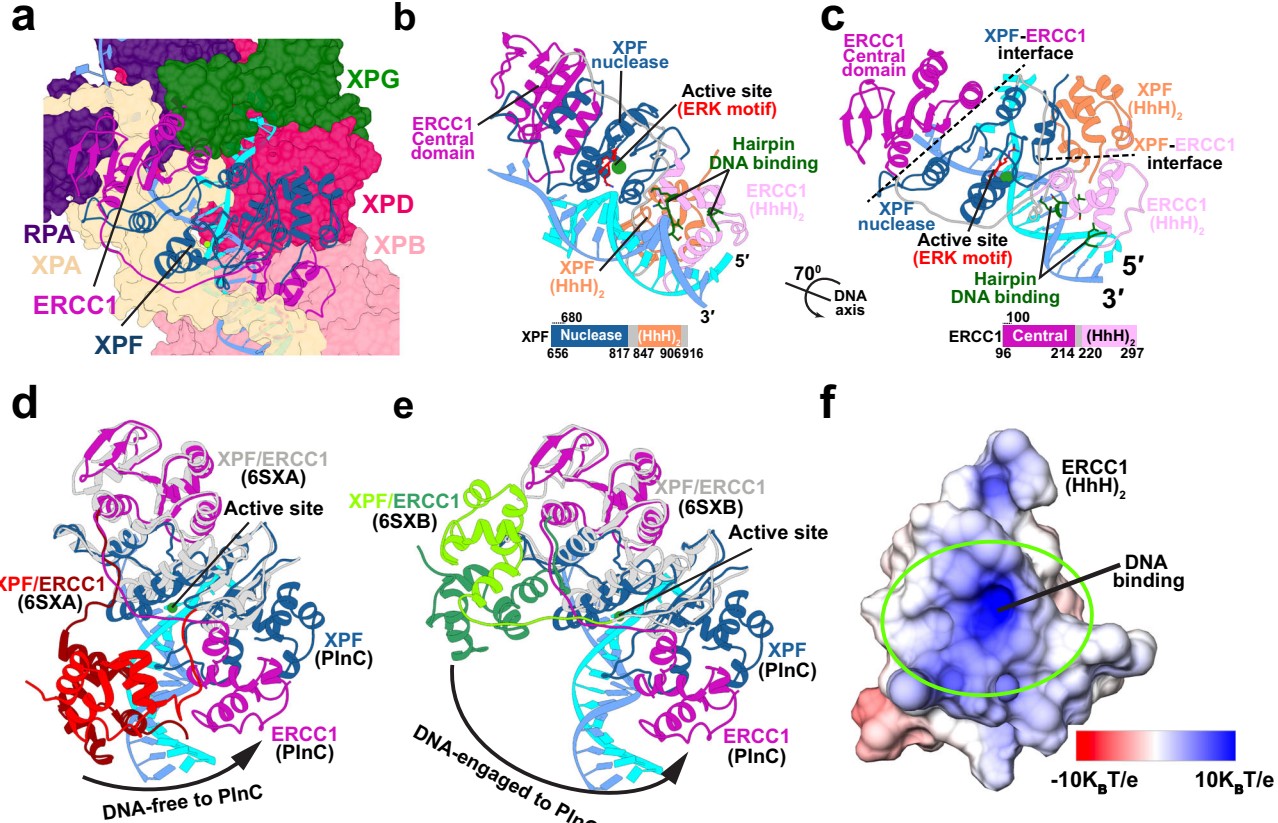

**Fig. 3 | XPF/ERCC1 is positioned for precise incision at the 5′ DNA junction.**
**a** View of XPF/ERCC1 at the 5′ junction. ERCC1 and XPF are depicted in cartoon representation and colored in magenta and dark blue, respectively. The damaged and undamaged DNA strands are shown in cyan and light blue, respectively. **b** View of the XPF/ERCC1 heterodimer interacting with DNA at 5′ junction. The ERK active site motif in shown explicitly and colored in red. XPF and ERCC1 are colored by domain. A schematic representation of the XPF and ERCC1 sequences, indicating the domains, is shown below. **c** View of the XPF/ERCC1 heterodimer rotated by 70° around the dsDNA axis. **d** Conformational shift of XPF/ERCC1 relative to the DNA-free XPF/ERCC1 structure (6SXA). **e** Conformational shift of XPF/ERCC1 relative to the DNA-engaged XPF/ERCC1 structure (6SXB). The XPF nuclease and ERCC1 central domains are overlayed for comparison. The XPF/ERCC1 (HhH)₂ domains are colored in red and dark red in the DNA-free structure and green and dark green in the DNA-engaged structure. ERCC1 and XPF in the PInC are colored in magenta and blue, respectively. **f** Electrostatics of the ERCC1 (HhH)₂ DNA-binding surface. The electrostatic potential is mapped onto the molecular surface and colored from red (negative) to blue (positive).

which is oriented in parallel to the ssDNA and directly contacts the DNA bases (Fig. 2d and e). This finding provides an unanticipated mechanism for XPG to sense the XPF/ERCC1 incision event at the 5′ junction. Importantly, such nuclease coordination mechanism would be mediated entirely by XPG, instead of relying on the flexible ssDNA to transmit the signal post-incision.

## XPG competes with XPC for binding to the 3′ DNA duplex and the p62 PH domain

In the PInC model, the XPG core occupies the position of XPC in the pre-unwound complex[15,16]. The fact that XPC and XPG both compete for binding to the 3′ DNA duplex and do not coexist in NER complexes[63,64], implies a mechanism must exist for XPG to displace XPC during the progression from lesion recognition to scanning and strand incision. Recruitment of TFIIH to the pre-unwound complex is essential for NER and requires XPC binding to p62's PH domain via a conserved acidic patch[60] (Supplementary Fig. 4). We identified a highly similar acidic patch in XPG (residues 151–164) on an outer loop of the XPG-helical bundle domain (Fig. 2d and Supplementary Fig. 4). We note this acidic patch aligns with residues 128–148 in the Rad2 sequence, which features multiple acidic segments competing for p62 binding[60]. If XPG and XPC use similar acidic patches to interact with the same interface on the flexible p62 PH domain, their binding would be competitive and mutually exclusive. The XL-MS data[46]

supports PH domain placement proximal to the XPG coiled-coil helices and the Fe-S domain of XPD (Supplementary Fig. 4). As the PH domain is key for XPC-TFIIH association[15], this observation provides a mechanism for handoff of the DNA substrate from XPC to XPG mediated by acidic patch-p62 interactions.

## XPF/ERCC1 is strategically positioned for incision at the 5′ DNA junction

A defining feature of the XPF nuclease is its modular domain architecture (Figs. 1b and 3b) that encompasses an SF2 helicase-like N-terminal domain, a central nuclease domain, and two C-terminal helix-hairpin-helix domains (HhH)₂. This flexible domain arrangement allows the nuclease to adapt its structure to accommodate diverse protein partners and substrate DNA. XPF functions as an obligate heterodimer with the ERCC1 protein, mediated by their (HhH)₂ domains. The ERCC1 central domain has high structural homology to the XPF nuclease domain. In our PInC model, the XPF/ERCC1 caps both sides of the 5′ DNA junction and is positioned for incision (Figs. 3a–c). Binding to duplex DNA at the junction is mediated mainly by the ERCC1 (HhH)₂ domains, which present a conserved electrostatically complementary DNA-binding surface (Fig. 3f and Supplementary Fig. 5). By contrast, the ERCC1 central domain binds ssDNA, associating mostly with the undamaged strand. The XPF nuclease domain binds directly at the ssDNA-dsDNA junction, facing the DNA duplex and placing the catalytic metal

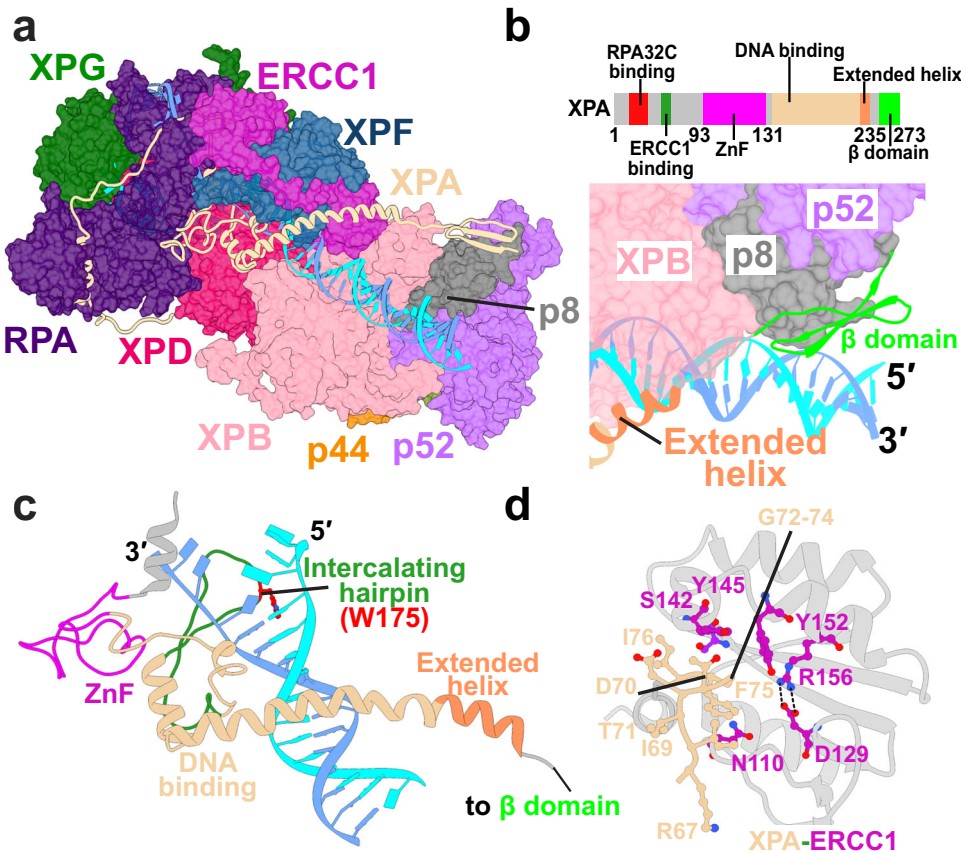

**Fig. 4 | XPA rigging interlaces XPF/ERCC1 with DNA, XPD, XPB, and RPA at the 5′ junction and is key for licensing the XPF incision. a** View of XPA within the PInC assembly. XPA is depicted in cartoon representations and colored in tan. **b** Zoomed-in view highlighting of the interface of XPA's β-domain and p8. TFIIH's p8, p52, and XPB subunits are shown in surface representation. A schematic of XPA's sequence colored by domain is shown above. **c** XPA interacting with DNA at the 5′ junction. XPA is colored by domains. The intercalating hairpin is shown in green and labeled. **d** Detailed residue interactions between the glycine-rich loop of XPA and the V-shaped groove of ERCC1. Residues are depicted in ball and stick representation and colored in purple and magenta.

3 Å away from the scissile phosphodiester bond. The Mg$^{2+}$ ion that affects catalysis is held by a GDX$_n$ERKX$_3$D active site motif[65], conserved among XPF family nucleases.

Achieving a catalytically competent state requires a major conformational shift of the XPF/ERCC1 complex relative to the known human XPF structures[43], DNA-free, and DNA-engaged XPF/ERCC1. While the XPF nuclease and ERCC1 central domains overlay well among the two structures and our PInC model, the tandem (HhH)$_2$ domains of both proteins are completely repositioned (Fig. 3d and e). The long flexible linkers connecting the (HhH)$_2$ domains to the nuclease core and the ERCC1 central domain make this feasible. We posit that completing this conformational switch is key for licensing the 5′ incision and requires interactions with other PInC constituent proteins (Fig. 3a). For example, the (HhH)$_2$ of XPF binds both XPB and XPD to ensure correct orientation of the ERCC1 (HhH)$_2$ for binding to the 5′ duplex. Additionally, the XPF nuclease and ERCC1 central domains form numerous contacts with XPA, which ensures stable association of XPF/ERCC1 within the PInC. Thus, XPA serves as an indispensable platform for the recruitment of XPF/ERCC1 to the pre-incision complex.

## XPA orchestrates recruitment of core NER factors to the PInC and licenses XPF for incision

XPA is a recognized master coordinator of nucleotide excision repair, controlling the assembly and coordinated exchange of core NER factors including XPC/Rad23B, RPA, TFIIH and XPF/ERCC1[5]. Despite its modest size, XPA spans the length of the PInC assembly, traversing and interlacing the surfaces of RPA, XPF/ERCC1, XPD, XPB, p8 and p52

(Fig. 4)[46]. In our model, XPA is strategically positioned at the 5′ edge of the NER bubble where it plays a key role in coordinating the repair process. XPA features a central DNA binding domain (DBD) flanked by largely disordered and flexible N- and C-terminal regions. The DBD inserts a β-hairpin and a tryptophan residue (W175) into the 5′ junction of the NER bubble (Fig. 4c). Thus, XPA provides the helicase pin to facilitate XPB's DNA-unwinding activity[46]. The DBD also includes a zinc-binding motif that is proximal to the ssDNA of the undamaged strand and forms key interactions to RPA[32], XPF/ERCC1[47] and XPD[46] (Figs. 1a and 4a). XPA's C-terminal end contains an extended helix that acts as a clamp on dsDNA and prevents the dissociation of the upstream duplex from XPB. This helical clamp concludes with an antiparallel β-sheet[16], which we modeled with AlphaFold2 (Fig. 4a and b). The interaction has been previously identified to anchor the XPA C-terminus to p52 and p8 but without structural detail[46]. The newly modeled XPA N-terminus is predominantly unstructured, except for a previously identified helix (residues 21–40)[32] that inserts into the RPA trimer core and binds the RPA32C domain (Fig. 5). The RPA32C-binding motif on XPA is situated >50 residues away from the DBD. Thus, it is key that we could model XPA's N-terminus connecting the helix and DBD in this particular RPA binding mode. Besides RPA, XPA mediates interactions with XPF/ERCC1 that are essential for NER progression. Specifically, our model replicates a critical interaction between the glycine-rich loop of XPA (residues 67–80) and a V-shaped groove of ERCC1[47], which serves as a validation point of our modeling (Fig. 4d). This 14-amino acid stretch of XPA has been shown biochemically as both necessary and sufficient for ERCC1 recruitment to the PInC. The intricate entanglement of XPA with

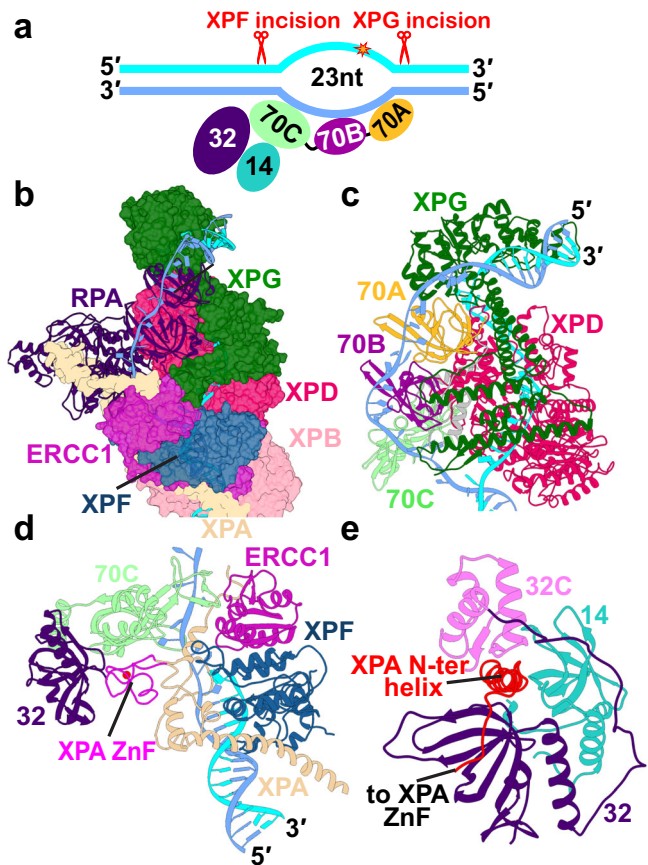

**Fig. 5 | RPA binds and protects the undamaged strand, engaging ssDNA with its 70A, 70B and 70C domains. a** Schematic representation of RPA's DNA-binding core with the ssDNA of the undamaged strand. The ssDNA is engaged by three OB-fold domains (70A-70B-70C). **b** View of RPA within the PInC assembly. RPA is shown in cartoon representation and colored in purple. **c** View of RPA70AB situated near the 3′ junction and interacting with XPG and XPD. RPA70AB is colored by domains. **d** View of the RPA trimer core engaging ssDNA at the 5′ junction. The XPA zinc finger is wedged between RPA70C and 32D. **e** A view of XPA's N-terminal XPA lodged between the RPA32C and RPA14 domains.

XPF/ERCC1, RPA and other key PInC components underscores its critical role in licensing the XPF incision and, more generally, in orchestrating NER progression.

## RPA matches the ssDNA length of the undamaged strand

A key component of the NER machinery, the single-strand DNA binding protein RPA protects the undamaged strand of the NER bubble. RPA features four DNA-binding OB-fold domains (RPA70A, 70B, 70C, and 32D) that modularly engage ssDNA in three attainable binding modes[48,66]. The first binding mode involves only the 70A and 70B domains and has an 8–10 nucleotide footprint on ssDNA. The second mode engages the 70C domain, increasing RPA's footprint to 20-24 nucleotides. Full RPA engagement covers up to 30 nucleotides. In our PInC model, RPA inserts itself between XPG, XPD, and XPA's DBD core. The length of the undamaged strand is surprisingly consistent with only three OB-fold domains (70A-70B-70C) specifically engaging ssDNA (Fig. 5). By contrast, previous NER models had assumed RPA binding with all four OB domains. There have also been conflicting models regarding the ordering of the RPA domains along the undamaged strand, some suggesting RPA70AB association near the 5′ junction and others—to the 3′ junction. Furthermore, interactions between the XPA DBD and RPA70AB have been reported as essential for the completion of the NER reaction[32]. In our model, RPA70AB

interacts with XPG near the 3′ junction (Fig. 5b and c), while the XPA zinc finger domain is positioned between RPA70C and RPA32D (Fig. 5d). The N-terminal XPA helix lodges between the RPA32C and RPA14 domains (Fig. 5b and e). This arrangement is consistent with the expected polarity of RPA on ssDNA and precludes direct binding of RPA70AB at the 5′ junction. To resolve this conundrum, we posit that RPA initially inserts only the 70A,B domains between the closely spaced junctions of the nascent NER bubble. The RPA70A,B module forms early contacts with the XPA zinc finger, explaining why these interactions are essential for NER. Subsequently, XPA moves with the 5′ edge of the expanding NER bubble while RPA70A,B remain bound to ssDNA near the 3′ junction. This process creates an opening on ssDNA for the insertion of RPA70C between RPA70A,B, and XPA and orients RPA32 and RPA14 near the 5′ junction.

## PInC's global motions and dynamic modules are key for dual incision coordination

The NER machinery undergoes a dramatic structural shift between the lesion scanning and strand incision phase. To reveal the impacts of this conformational switching on the functional dynamics of the NER complexes, we performed microsecond-timescale molecular dynamics simulations of PInC, LSC and apo-TFIIH. We compared their relative flexibility by mapping computed B-factors from MD onto the structural models (Supplementary Fig. 6 and Supplementary Note 2). TFIIH mobility is markedly reduced in the PInC compared to LSC, yielding exceptionally low B-factors (Supplementary Fig. 6c and f). We observe a ridge of stability that extends from the XPG core and encompasses TFIIH's XPD, XPB and p44 subunits, most of RPA, XPA's DBD, and XPF/ERCC1. Notably, in the LSC the opening/closing dynamics of XPD's Arch and Fe-S domains is preserved (Supplementary Fig. 6e), which is essential for lesion scanning. By contrast, this dynamics is abolished in the PInC (Supplementary Fig. 6f). Thus, the NER machinery progressively loses mobility from DNA-unwinding to lesion scanning and strand incision. Loss of XPD's residual mobility converts PInC into a rigid platform for the assembly of XPF/ERCC1 and XPG.

To analyze and visualize the concerted global motions of the PInC, we relied on two methods—dynamic network analysis[67,68] and principal component analysis (PCA)[69]. Using covariance data from the MD simulations, network analysis partitions PInC into communities representing the assembly's dynamically independent structural modules. Once we identify PInC's moving parts, PCA helps us determine the directionality of their motions (Supplementary Fig. 7).

Network analysis identifies 22 dynamic communities, which are color-coded and mapped onto the PInC structure (Figs. 6a and 7). The edge betweenness graph (Fig. 6b) encodes the magnitude of allosteric communication between pairs of communities. Notably, XPG separates into four communities (H, M, R, S). While XPG's I- and N-domains are structurally entangled in forming the nuclease core, the M and R communities encompassing XPG's core do not separate by domain. Instead, the catalytic core splits along the central β-hairpin leading into the XPG active site. Community M, carrying both the H2TH and α12b motifs, is sculpted to bind two turns of duplex DNA. Predictably, the motions of the 3′ DNA duplex closely mirror community M's dynamics (Supplementary Fig. 7). By contrast, community R, containing the hydrophobic wedge and adjacent helices, associates with the ssDNA side of the 3′ junction. Importantly, XPG's coiled-coil helices form a single dynamic module with XPD's Arch domain and the ssDNA passing through XPD (community H). Thus, helical arch dynamics is decoupled from the XPG core (Fig. 6b), allowing the coiled-coil to serve as a rigid cap on XPD's DNA-binding groove. The two long helices effectively block the opening/closing dynamics of the Arch and Fe-S domains required for ssDNA translocation. Consistently, the first four PCA modes (Supplementary Fig. 7) show the Arch, Fe-S, and coiled-coil helices predominantly moving concertedly with the same directionality. We also observe a strong edge between communities H and

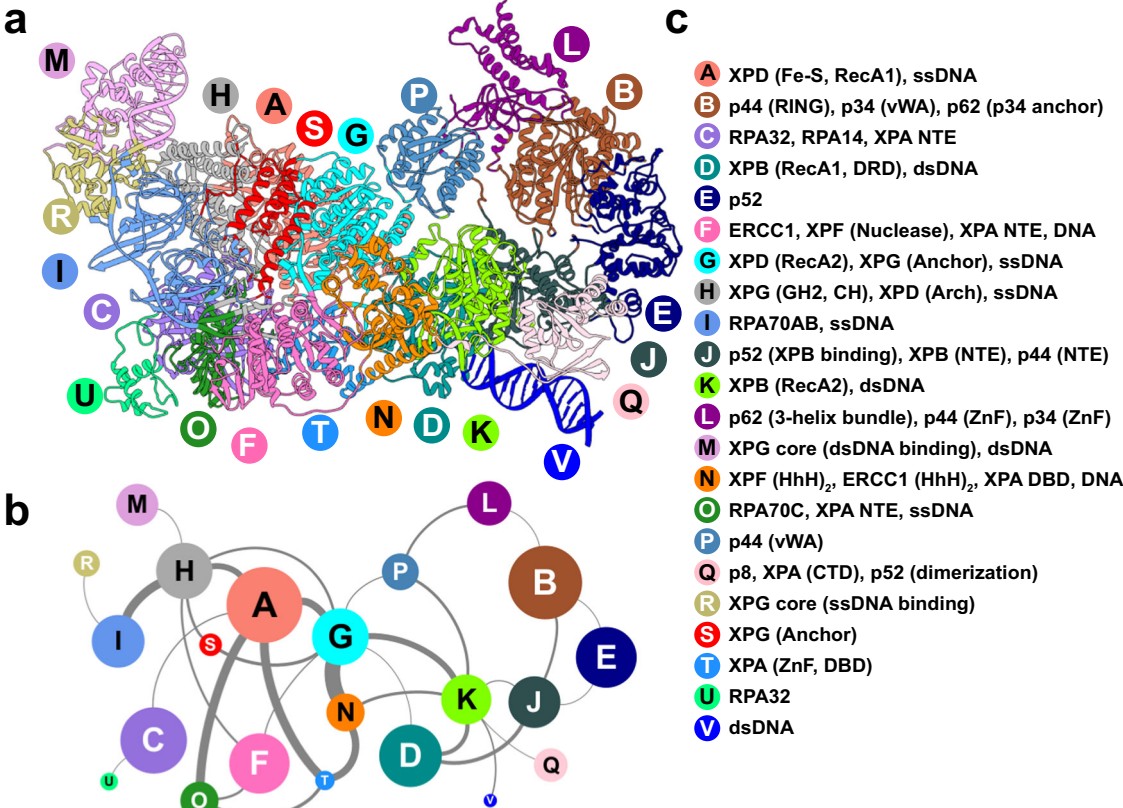

**Fig. 6 | Network of dynamic communities underlie PInC's functional dynamics.** **a** Communities identified from dynamic network analysis that transcend subunit divisions. **b** Graph of allosteric communication among communities. Nodes are sized by the number of residues in each community. Edge thickness represents the magnitude of dynamic communication between communities (betweenness). **c** Labels identifying the domains or structural elements participating in each dynamic community.

A, suggesting tight coupling of the XPG helical arch with XPD's Arch, Fe-S, and RecA1 domains. We also note the plug region of XPD – a functionally important, but also a highly mobile and conformationally variable region of PInC[70]. Modeling XPD with AlphaFold2 shows the plug to be consistent with the conformation found in the preinitiation complex (Supplementary Fig. 8a). However, the low pLDDT values suggest low fold prediction confidence for this region. Independent simulations of XPD in isolation also show that the plug is mobile and undergoes partial unfolding (Supplementary Fig. 8b and c). Intriguingly, the XPG-anchor domain forms a separate community (S), which is dynamically coupled to community G that encompasses mainly the RecA2 domain (Fig. 6b). Through communities G and A, the XPG-anchor serves as an essential link to the communities encircling the 5′ DNA junction: F (ERCC1 central, XPF nuclease domains, XPA-NTE), T (XPA DBD and ssDNA) and N (XPF and ERCC1 (HhH)₂, XPA DBD and dsDNA). Thus, our network analysis and PCA results corroborate our postulated model for the XPG-Arch domain acting as a sensor of the 5′ incision and as a bridge between XPF/ERCC1 and the XPG core.

At the 5′ DNA junction, two large communities dominate—1) community F encompassing ERCC1's central and XPF nuclease domains; and community N, which includes the XPF and ERCC1 (HhH)₂ and XPA's extended helix domains. A segment of XPA's DBD and intercalating β-hairpin form their own smaller community with ssDNA (community T). The extent to which XPA is entangled with other core NER proteins (RPA, ERCC1, XPF, and p8) is remarkable, with XPA contributing to six distinct dynamic modules (Fig. 6a, communities C, O, F, T, N, and Q). In essence, XPA serves as a structural adhesive to link RPA with ERCC1 and XPF and to organize them at the 5′ junction. This finding supports and extends current understanding of XPA's multifaceted function in NER: 1) XPA acts as a clamp on dsDNA to facilitate NER bubble expansion; 2) recruits and positions RPA at the NER bubble; and 3) serves as a platform to assemble XPF/ERCC1 within the PInC. Unlike XPA, RPA subdivides mostly by domains, highlighting its modular architecture. The RPA 70A and 70B domains have motions that are well correlated. Predictably the domains form a single dynamic module (community I). By contrast, the 70C domain is less coupled to the 70A, 70B pair and forms its own community O. Intriguingly, community O is strongly coupled to community T, which incorporates XPA's zinc-finger domain (Fig. 6b). The zinc-finger is key for stabilizing RPA at the 5′ DNA junction. RPA14 and RPA32 form a single dynamic module (community C), which also includes XPA's N-terminal helix.

## Disease mutations cluster at critical junctures of the PInC dynamic network

The pre-incision complex is arguably the most critical NER intermediate. Correspondingly, disruption of this intricate molecular machine by mutations gives rise devastating human genetic diseases. Specifically, there are three diseases linked to defects in PInC constituent proteins, including TFIIH: xeroderma pigmentosum (XP), Cockayne-syndrome (CS), or trichothiodystrophy (TTD). XP, TTD, and XP/CS are autosomal recessive genetic disorders. Patients are often compound heterozygotes having two distinct mutations in each allele. Both alleles contribute to the expressed phenotype with combined phenotypes possible (e.g., XP/TTD, XP/CS)[71]. In general, XP mutants are GG-NER defective, TTD mutants cause partial transcription defects, XP/TTD mutations exhibit both defects and XP/CS mutations are defective in both GG-NER and TC-NER.[6–8,72,73] To link molecular features to disease phenotypes, we mapped missense disease mutations (Fig. 7, Supplementary Table 1, and Supplementary Movie 2) onto our integrative PInC model. Mutations are

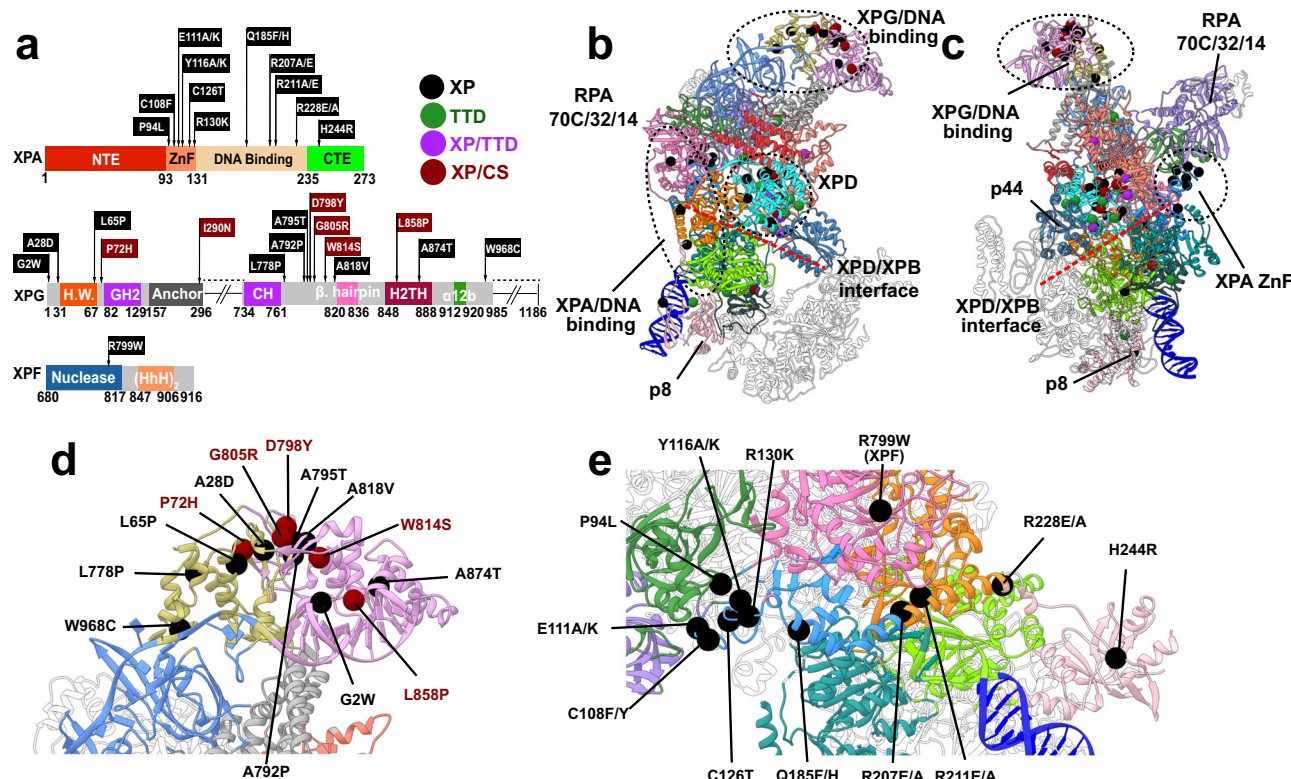

**Fig. 7 | Human disease mutations mapped onto the PInC model are shown to localize at key community interfaces. a** Missense disease mutations mapped onto the sequences of XPG, XPA, and XPF. Disease phenotypes are labeled as XP, XP/CS, TTD, and XP/TTD and identified by symbols and color. **b** Map of human disease mutations (spheres) onto the PInC structure (anterior view) shown in cartoon representation and colored by community. The three primary clusters of mutations associated with XPG, TFIIH, and XPA are outlined with black dashed lines. **c** Map of human disease mutations onto the PInC structure shown in posterior view. **d** Close-up view of mutations within XPG. **e** Close-up view of mutations within XPA and XPF.

distributed in an irregular spatial pattern with three major clusters emerging: 1) the XPG catalytic core harbors 13 mutations; 2) 34 are confined within TFIIH's XPD, XPB, and p8 subunits; and 3) 11 fall within XPA. The XPG anchor and XPF nuclease core each carry a single mutation. With all TFIIH-resident mutations previously annotated[27,37], we focus our analysis on the XPG, XPF, and XPA clusters. The XPG nuclease core splits into two dynamic communities (M and R) with the community interface particularly rich in disease mutations (A28D, A795T, A792P, A818V are XP type; G805R, L858P, D798Y, P72H are XP/CS type)[41]. Many of these involve substitutions of hydrophobic for polar or charged residues. This not only disrupts the local dynamics close to the XPG active site but also affects the stability of the central β-sheet bridging the M and R communities of the nuclease core. To determine the effect of mutations on XPG protein stability, we used the Rosetta ddG protocol (Supplementary Table 1)[74]. There is a striking difference in the ddG scores between XPG and XPA resident mutations (Supplementary Fig. 9 and Supplementary Movie 3), suggesting XPG mutants act largely by protein destabilization. By contrast, XPA mutants act by disrupting key interactions and dynamics at community interfaces. Notably, mutations within the XPG catalytic core produce large computed ddG scores and are thus predicted to be highly destabilizing (Supplementary Fig. 9 and Supplementary Movie 3) consistent with mutant expression levels[41]. XPG also contains mutations that are internal to communities M and R: 1) G2W, W814S, A814T, and W968C directly affect DNA binding; 2) L778P and L65P alter the backbone of key helices, disrupting the nuclease fold. Intriguingly, the single disease mutation located outside the XPG core is I290N[8], which disrupts a hydrophobic cluster next to the β-hairpin (residues 265–296; Fig. 2e) at the edge of the XPG anchor domain. Proximity to the 5′ junction ssDNA and the XPF/ERCC1 interface suggests that the β-hairpin is functionally significant. We posit that this

structural element may be key for sensing incision at the 5′ edge of the NER bubble.

All XPA disease mutations are XP phenotype and are distributed across the entire length of the protein. Notably, XPA disease mutations lie primarily at community interfaces (Fig. 7 and Supplementary Movie 2). Six mutants cluster within the zinc-finger domain (P94L, C108F, C126T, E111A/K, Y116A/K, R130K) (Figs. 7c and f). Structural disruption of the zinc finger impairs XPA–DNA binding, impacting NER activity[75]. The zinc-finger domain is also the linchpin of the RPA70C interface with XPA's DBD, bridging dynamic communities O, C, and T. Destabilization or dynamic disruption of this interface could abolish the interaction of XPA and RPA, which is critical for NER. The C-terminal domain of XPA harbors a single XP mutant (H244R), which affects XPA–TFIIH association and decreases NER activity. From our model, we note the proximity of H244 to dsDNA. Switching the residue to an arginine increases DNA binding at the expense of weakening the interface between XPA's C-terminal β-sheet and TFIIH's p8 subunit. The XPA R228A/E mutation lies in the extended helix of XPA at the interface of communities Q and N. Disruption in that region not only affects DNA binding but could also impact the dynamic module formed by XPA and ERCC1's (HhH)$_2$ or its interactions with XPB. The remaining XP mutants in XPA lie in the DBD. Strikingly, all three are situated at dynamic community interfaces (Q185F/H between communities T and G; R211E/A and R207E/A between communities N and T) and are poised to disrupt DNA binding or interfere with the complex entanglement of XPA with XPF and ERCC1 at the 5′ DNA junction (Fig. 7e and f). By contrast, the lone mutation in XPF (R779W) is internal to community N but lies at the interface between XPF nuclease and ERCC1 central domains and is C-capping an ERCC1 helix. Mutation to tryptophan would disrupt the helix and the XPF–ERCC1 interface to affect recruitment of XPF/ERCC1 to TFIIH.

## Discussion

NER complexes are amazingly dynamic protein machines that actively reshape themselves and self-regulate to achieve precise lesion removal from genomic DNA. By synthesizing cryo-EM and XL-MS data with advanced computational modeling, we built a practically complete hybrid model of the NER pre-incision complex and analyzed its functional dynamics. The results support a structural basis for XPF and XPG nuclease licensing and coordination of dual incision in NER.

In early NER, the XPC/Rad23B/CETN2 complex first detects lesioned DNA, recruits TFIIH in an open apo-like conformation, and stabilizes a nascent NER bubble. At this stage, TFIIH's XPD subunit is distal from DNA while the XPB subunit acts as a translocase and unwinds DNA downstream of the lesion site. TFIIH's CAK module presents an obstacle to further NER progression and is removed upon subsequent XPA recruitment to the 5′ end of the NER bubble. This conformational switch involves displacement of CAK's MAT1 subunit by the N-terminus of XPA. It has been previously proposed that MAT1 could serve as XPB-XPD spacer, and its removal could allow XPD to approach DNA[15,37]. Additionally, XPA stimulates XPB unwinding and orchestrates initial recruitment of RPA. In turn, RPA binds and protects the undamaged strand between its 70A and 70B domains. Throughout this process, XPC remains bound at the upstream DNA duplex.

Our computationally informed mechanism (Fig. 8), sheds light on the late-stage reorganization of the NER protein machinery from lesion scanning to dual incision. Once the NER bubble reaches a critical size, XPD threads the lesioned DNA strand through its DNA-binding groove, triggering closure of TFIIH and a collapse of the XPD–XPB spacing. The ensuing close association of XPD, XPB, and p44 blocks XPB's unwinding activity[37], leading to the formation of the LSC. Thus, the action of XPB and XPD in NER is sequential and strictly coordinated. During the lesion-scanning phase, XPD reels in ssDNA toward the 5′ junction, unwinding the upstream DNA duplex. When the lesion becomes blocked in the narrow space between XPD's Fe-S and Arch domains, DNA unwinding stops. This event triggers assembly of the PInC. Intriguingly, the helicase pins assisting strand separation at the edges of the NER bubble are not internal to the two translocases. Instead, they are provided by other proteins (XPG and XPA), ensuring XPB and XPD unwinding activities are optimally efficient only in the context of a fully assembled NER machinery.

In the next stage of NER, XPD activity is disabled, allowing the XPG and XPF/ERCC1 nucleases to assemble on the circularized and rigidified TFIIH scaffold. Concomitantly, RPA expands its footprint on the undamaged strand to include the 70C domain. XPG replaces XPC at the 3′ edge of the NER bubble, perhaps as early as the lesion scanning stage as there is evidence XPG could stimulate XPD activity[46,54]. The newly identified XPG anchor domain is critical for this handoff, which involves XPG binding to XPD, competing with p62's XPD anchor and BSD2 domains, and displacing XPC in its interaction with the PH domain of p62. Remarkably, XPG spans the entire length of the NER bubble and inserts a β-hairpin from the XPG-anchor into the 5′ junction close to the XPF/ERCC1 interface. Thus, our findings support an unexpected mechanism for XPG to sense the action of XPF and coordinate the dual incision of the lesioned strand. XPG is recruited early to the PInC, but XPF incision occurs first. Importantly, mere recruitment of XPF is insufficient for DNA cleavage. We posit that licensing of the XPF incision requires conformational switching to achieve the observed complex entanglement of XPA DBD, ssDNA, XPF and ERCC1 at the 5′ junction. Similarly, XPG nuclease licensing is triggered by termination of XPD unwinding that may involve tilting of the XPG core (see alternative XPG model). Reorientation of the XPG core could allow interaction with XPD's Arch domain to stimulate the 3′ incision. The final stage of NER involves PInC disassembly and departure of TFIIH with the excision product. Gap-filling DNA synthesis ensues and restores DNA to its original condition.

Collectively, these findings elucidate the structure and dynamics of a critically important state of the NER machinery – the pre-incision complex. The practically complete model and computational analyses yield key mechanistic insights into PInC's assembly and regulation, the structural basis of XPF and XPG nuclease coordination, and the licensing of the NER dual incision.

Linking molecular mechanisms to disease phenotypes is a grand challenge for structural biology. This challenge is often unmet as it requires knowledge of dynamic conformations and assemblies that resist purely experimental approaches. Our integrative methods and results provide a framework for meeting this challenge and for designing future experiments to uncover the intricate molecular choreography of global genome NER. Our dynamic network models powerfully elucidate the etiology of devastating human genetic syndromes. Notably, we find that XP and XP/CS disease mutations cluster at key interfaces of PInC's dynamic communities, impacting NER protein stability, functional dynamics, DNA binding, nuclease licensing, and/or community integrity.

## Methods

### Model building

To construct a model of the pre-incision complex (PInC), we systematically examined the cryo-EM structures and densities of human apo-TFIIH[22], TFIIH/XPA/DNA[46], and XPF/ERCC1[43], the NMR structure of XPA-ERCC1[47], and the X-ray structures of the XPG catalytic core[38] and RPA-ssDNA (RPA70, RPA32, and RPA14)[48]. The TFIIH/XPA/DNA structure[46] (PDB ID: 6RO4 and EMDB accession code: EMD-4970) was the starting point for model building. The PInC hybrid model has an NER bubble size of 23 nucleotides, matching the 27-nucleotide optimal length of the excision products and the XPF and XPG incision patterns. Sources of experimental structural information used in constructing the integrative model are summarized in Supplementary Fig. 10 and Supplementary Table 2. Regions modeled with AlphaFold2 are shown in Supplementary Fig. 11 with pLDDT scores mapped onto the structures. Additionally, maps showing geometric and electrostatic complementarity as well as conservation across the newly modeled PInC interfaces are shown in Supplementary Fig. 12.

FEN1 shares 30% sequence identity with the XPG catalytic core[76,38] (PDB ID: 6TUR, 6TUW, and 6VBH). Thus, we modeled DNA-bound XPG based on the human FEN1/DNA X-ray structure[61] (PDB ID: 5UM9). XPG positioning into the hybrid model was based on existing XL-MS data[22]. In addition, positioning of the XPG core required placement of the 3′ DNA junction 8 nucleotides away from the expected position of the DNA lesion near XPD's His135 residue. The two XPG gateway helices (GH1 residues 33–41 and GH2 residues 82–129) and the capping helix (CH, residues 734–763) were predicted with AlphaFold2[49,50] and positioned in the gap between XPD's Arch and Fe-S domains in accordance with the crosslink data[22]. The XPD-anchor domain (residues 157–296) was predicted by AlphaFold2 and fitted into the TFIIH/XPA/DNA cryo-EM density. The loop connecting GH2 and the XPD-anchor was built with Modeler[77].

To model XPF/ERCC1, we used the cryo-EM structures of XPF/ERCC1[43] (PDB ID: 6SXA and 6SXB). We first docked the XPF nuclease domain to the 5′ junction. The catalytic metal was oriented 3 Å away from the scissile phosphodiester bond. $Mg^{2+}$ ion coordination was based on the *Aeropyrum pernix* SNF2 structure[65] (PDB ID: 2BGW). A water molecule was placed between $Mg^{2+}$ ion and the DNA backbone phosphate group. The ERCC1 $(HhH)_2$ domain was oriented to interact with the ssDNA through two DNA hairpins based on the 6SXB structure. The long linkers from the ERCC1 central domain to the ERCC1 $(HhH)_2$ (residues 214–230) and from the XPF nuclease domain to the XPF $(HhH)_2$ (residues 817–847) were built with Modeler. The SF2 helicase-like N-terminal domain of XPF was omitted from the hybrid PInC model due to lack of sufficient structural or biochemical restraints.

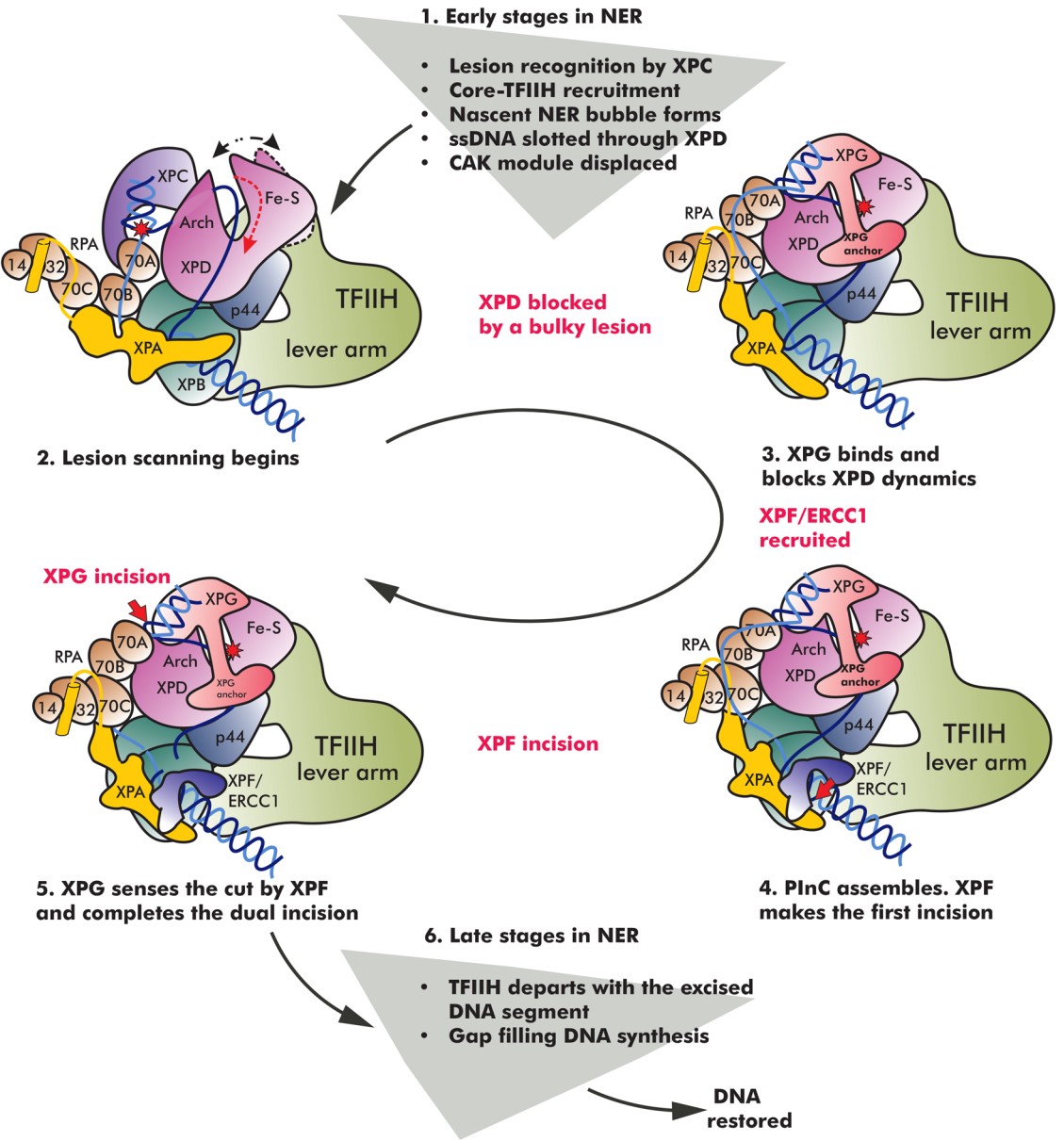

**Fig. 8 | NER protein machinery undergoes a dramatic structural reorganization from lesion recognition to lesion scanning and strand incision.** The schematic represent key steps in the NER pathway—XPC lesion recognition, NER bubble extension, XPD-mediated lesion scanning, PInC assembly and dual incision of the damaged DNA segment, gap-filling synthesis, and DNA restoration. Core NER factors are shown in cartoon representation and color-coded. The position of the lesion is indicated by a red star. Red dashed arrows show the direction of ssDNA movement during the different stages of NER. White dotted arrow denotes the opening/closing dynamics of XPB during bubble expansion. Black dotted arrow denotes the opening/closing dynamics of XPD during lesion scanning. A red cross denotes the blocking of a lesion inside XPD and damage verification. Red arrows indicate the incision points on DNA by XPF/ERCC1 and XPG, respectively.

To model RPA, we used following X-ray structures: *Ustilago maydis* RPA/ssDNA[48] (PDB ID: 4GOP), yeast RPA/ssDNA[78] (PDB ID: 6I52), and human RPA[79,80] (PDB ID: 1JMC and 1L1O). The RPA70AB/ssDNA complex was modeled by superimposing the yeast RPA/ssDNA structure[79] (PDB ID: 6I52) onto the human apo-RPA 70AB[78] (PDB ID: 1JMC). Within PInC, only RPA70A, 70B, and 70C can engage DNA due to the size of the NER bubble. RPA70AB was placed close to the 3′ junction where it interacts with XPG. We reoriented RPA70C to bind ssDNA near the 5′ junction. The RPA70C/ssDNA was modeled by aligning the *Ustilago maydis* RPA/ssDNA structure[48] (PDB ID: 4GOP) with the human trimer core structure[1] (PDB ID: 1L1O). The orientation of RPA32D and RPA14 follows from the placement of the RPA70C module as they are all connected, forming the trimer core (70C/32D/14).

To model XPA, we used the following structures: the cryo-EM TFIIH/XPA/DNA structure[46] (PDB ID: 6RO4), the NMR structure of XPA/ERCC1[47] (PDB ID: 2JNW), and the human X-ray structure of RPA32C/Smarcal1 N-terminus[81] (PDB ID: 4MQV). The XPA N-terminal extension (residues 1–103), which includes the RPA32C binding helix (residues 22–40), and the C-terminal extension (β-domain) (residues 235–273) lacked known structural homologs and were modeled using AlphaFold2. The β-domain was fitted into the TFIIH/XPA/DNA density. To position XPA's N-terminal helix (residues 22–40) we used the X-ray structure of RPA32C/Smarcal1 N-terminus.

To assemble the complete PInC model, we also modeled loop regions of TFIIH's core subunits (XPB, XPD, p44, p34, and p52) into the TFIIH/XPA/DNA density.

**Molecular dynamics**
Molecular dynamics simulations of PInC complex in the presence and absence of a CPD lesion were performed on the Summit machine of the

Oak Ridge Leadership Computing Facility. The systems were set up with the TLeap module of AMBER[82] and solvated with TIP3P water molecules[83]. To balance the overall charge of each system, we used Na+ counterions. Extra Na+ and Cl− ions were introduced to produce 150 mM salt concentration as needed to mimic physiological conditions. We then used the NAMD code to carry out energy minimization for 5000 steps with positional constraints imposed on the protein and DNA backbone atoms. NVT simulations were used to gradually bring the temperature of the systems to 300 K over a period of 100 ps. During the NVT run positional restraints ($k = 10$ kcal mol$^{-1}$Å$^{-2}$) were imposed on all heavy atoms of PInC. Equilibration was continued for another 5 ns in the NPT ensemble while the restraints were gradually released. Production simulations were carried out in the NPT ensemble (1 atm, 300 K) for 1μs for each complex. The particle mesh Ewald (PME) method was employed to compute the long-range electrostatic interactions. The r-RESPA multiple-time-step method[84] was used with a 2-fs timestep for bonded interactions, a 2-fs timestep for short-range non-bonded interactions, and 4-fs timestep for long-range electrostatic interactions. A short-range non-bonded interaction cutoff of 10 Å and a switching function at 8.5 Å were used for the simulations. All covalent bonds to hydrogen atoms were constrained using the SHAKE method. The simulations were performed with the NAMD 2.14 code[85] and the AMBER forcefields: Parm14SB[86] and OL15[87]. All figures were generated with UCSF Chimera[88].

## Covariance-based community network analysis

Covariance-based community network analysis[67,68] was performed on the PInC trajectories. Network analysis is a graph-theoretical method, which maps the PInC assembly onto a protein network graph wherein residues are represented as nodes and edges connect contacting residues. Two non-adjacent residues are considered in contact if they are within 4.5 Å for 75% or more of the trajectory. The MDTraj package[89] was used to obtain contact maps from the MD ensembles. Edges are weighted by residue-residue covariances given by: $w_{i,j} = -\ln(|c_{i,j}|)$, where $c_{i,j}$ are pairwise correlation coefficients. Partitioning this graph into strongly connected components with the Girvan-Newman algorithm[90] defines dynamic communities, which are PInC's dynamically independent functional modules. Edge betweenness among communities then recapitulates allosteric communication within the assembly.

## Principal component analysis

PCA is a dimensionality reduction technique, which diagonalizes the residue-residue covariance matrix from the simulation trajectories, yielding corresponding eigenvectors (principal modes) and eigenvalues (mean square fluctuations). The first few principal modes recapitulate the functionally significant large-scale motions of the PInC assembly. PCA was performed using the CPPTRAJ module in AmberTools16[91].

## Rosetta protein stability analysis

XP and XP/CS disease mutations in XPA, XPG, and XPF were assessed for their impact on protein stability using the Rosetta Cartesian ddG protocol[74]. The wild type (WT) structure of the PInC complex was relaxed in Cartesian space using the Rosetta FastRelax protocol. Mutations were then introduced and the FastRelax protocol used to repack the side chains within 6 Å of the mutation site. The protein backbone within 3 residues of the mutation site is also allowed to readjust. The ddG value were determined by the Rosetta score differences between the relaxed mutant protein and the relaxed WT protein for each of the evaluated missense mutations.

## Reporting summary

Further information on research design is available in the Nature Portfolio Reporting Summary linked to this article.

## Data availability

The models of the PInC complex have been deposited in the PDB-dev database with accession codes: PDBDEV_00000373, 9A88, and PDBDEV_00000374, 9A89. The final configuration of the PInC molecular dynamics trajectory is provided as a plain text file pre-incision-complex-final-MD-configuration_PDB.txt in PDB format as Supplementary Data 1 file. PDB cccession codes of all the publicly available datasets used in the study: 6RO4, 6TUR, 6TUW, 6VBH, 5UM9, 6SXA, 6SXB, 2BGW, 4GOP, 6I52, 1JMC, 1L1O, 2JNW, and 4MQV.

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

## Acknowledgements

We thank Wei Yang (NIDDK) and Orlando D. Schärer (UNIST, South Korea) for insightful discussions. This work was supported by the National Institute of General Medical Sciences grant R35GM139382 (I.I.), the National Institute of Environmental Health Sciences grant R01 ES032786 (I.I., S.E.T and C-L.T.), the National Science Foundation grant MCB-2027902 (I.I.), and NCI grants P01 CA092584 (J.A.T., I.I., S.E.T. and C-L.T.), R35 CA220430 (J.A.T.) and core funding by BAS/1/1002-01-01 to S.M.H. An award of computer time to I.I. was provided by the INCITE program. This research also used resources of the Oak Ridge Leadership Computing Facility, which is a DOE Office of Science User Facility supported under Contract DE-AC05-00OR22725.

## Author contributions

I.I. directed the study. J.Y., C.Y., S.E.T, and I.I. contributed to the design of the study. J.Y., C.Y., T.P. and L.B. performed model building and molecular simulations of the models. J.Y., C.Y., T.P., L.B., S.E.T., C-L.T., S.M.H., J.A.T and I.I. analyzed the data. S.E.T., C-L.T., S.M.H. and J.A.T provided critical comments during the analysis of the data. All authors discussed the results and were involved in the editing of the manuscript.

## Competing interests

The authors declare no competing interests.
