## [Peer Review File · Nature Communications]

Molecular architecture and functional dynamics of the pre-incision complex in nucleotide excision repairEditorial Note: Parts of this Peer Review File have been redacted as indicated to remove third-party material where no permission to publish could be obtained.

REVIEWER COMMENTS

Reviewer #1 (Remarks to the Author):

Yu et al propose a molecular model of the human nucleotide excision repair complex by synthesising available structural information (chemical crosslinks, experimental structures), addition of AlphaFold2 modelling and molecular dynamics simulations. They use the obtained model to propose a lesion recognition mechanism and interpret distinct classes of human disease mutations in various subunits. The overall theme of the research, the architecture and mechanism of the highly dynamic and complex nucleotide excision machinery is a timely topic of broad and general interest. A validated, high-quality model of the machinery is a valuable resource for the field and beyond. The computational tools are reasonably well described and are state of the art. The paper is generally well written and covers a lot of ground.

The authors derive interesting conclusions from the model with respect to lesion recognition and cleavage as well as disease mutations.

I have two main points regarding the work in this manuscript that need to be addressed before publication.

1) No independent experimental validation. While the molecular model is consistent with available information, it was not validated by a model-derived testable hypothesis. I am aware that this is quite a challenge for such a complex system and perhaps beyond reach at this stage. However, perhaps the authors could at least perform a robust interface analysis that provides orthogonal measures for the quality of experimentally known (old) and modelled (new) interfaces (stereochemistry, packing quality, chemical complementarity etc).

2) I think it is ok to perform the in silico analysis of the disease mutations on top the (unvalidated) model, however I find the proposed repair mechanism too detailed and speculative at this stage without experimental validation or new experimental data. It would be ok though to add a schematic panel as a simplified conclusion from the model.

Other points for revising the manuscript:

1) It would be tremendously helpful to provide in the supplement an overview figure highlighting on their pre-incision complex model, which part is derived from experimental structures, which parts are modelled by which computational models. For instance a replica of Fig 1 but showing the source of information for each subparts.d

2) The depiction of crosslinks in Fig. 2 is not ideal since one only sees crosslinked residues but not the path of the crosslinks. Perhaps it gets to confusing, but I suggest to add a panel, e.g. in the supplement that shows the paths of the crosslinks on the structures. Furthermore, a bit more detail is needed here, to what extend do crosslinks agree with the structure at a given cutoff for the type of crosslinker used.

3) The color scheme for the disease mutations in the movies is not very clear, the white or light pink spheres are practically invisible and what stand out are only the dark blue or black spheres. Is this intended? If not I suggest a scheme based on variations on hue but keeping saturation similar so that all proposed groups are equally visible?

4) XPD is a 5' -> 3' helicase correct? I am confused by the sentence on p16 "XPD reels in ssDNA in the 3'to 5' direction", please clarify or reformulate.

Reviewer #2 (Remarks to the Author):

Yu et al. present a modeling study on the handover of the XPC/centrin2/Rad23B complex after initial lesion recognition and the handoff to TFIIH as central factor in the pre-incision complex responsible for lesion verification and positioning of the nucleases as well as priming them for incision. This study is a follow up on their previous study from 2023 and now includes additional proteins and steps in the NER process. The authors utilized a combination of cryo EM and crystal structures, interpreted additional densities in cryo EM maps, and modeled those based on existing additional data. The modeled complexes were then subjected to further computational analyses to display global motions, dynamic communities and towards the impact of disease related mutations. Overall, the study provides interesting perspectives and possibilities on how a pre-incision bubble may be formed, how the size of the bubble is determined, and how the nucleases are positioned for incision. The authors describe potential interactions and possible molecular crosstalk between subunits like XPG and XPD, XPA and XPF/ERCC1, XPG and p62. They also provide a plausible model for RPA protecting the undamaged DNA strand.

This is an interesting study provoking thoughts how core TFIIH may function and could crosstalk to other NER factors. However, there are some serious concerns which prevent a recommendation for publication in Nature Communication:

- Overall, the study is highly speculative. None of the proposed complexes and most importantly protein-protein interactions and protein-nucleic acid interactions were further analyzed or verified by biochemical experiments.

- The authors claim to provide a model leading from initial damage recognition to damage verification. However, they entirely ignore the required dissociation of the CAK complex. This is an absolutely necessary step in the NER process. The detailed description of the individual steps within the manuscript implies that they have been all addressed which is clearly not the case.

- The additional modelling performed by the authors based on the map of 6ro4 is very biased towards XPG replacing p62 at the anchoring position. The crosslinking data from previous studies support that XPG can bind in that area but there is also data supporting the presence of p62. Inspection of the residual density modelled in this study shows no additional density that supports the presence of XPG rather than p62 thus making their interpretation even less likely. Furthermore,

the crosslinking data do not show which kind of particles went into the final map of the cryo EM structure. It is thus unclear if XPG or p62 are present in the map. Furthermore, in an earlier study by Barnett et al. NAR 2020 (7ad8) p62 was built into the residual density which should be discussed here. The authors can argue that this is a mutual binding site for the anchor, it is however highly unlikely that this is exclusively XPG or XPG at all, since there is clear evidence for the presence of p62 and virtually none for the presence of XPG. A detailed map of the crosslinks mentioned with respect to p62 and XPG in that region of XPD would also be appreciated and seems necessary.

- The here provided coordinate model displays the XPD plug region, that has been shown to be of functional importance, modeled very closely to the apo form of TFIIH that still contains MAT1. In 6ro4 this part of the XPD molecule is disordered and Kokic et al. argue that releasing the plug is activating XPD helicase. This seems to be contradictory to the modelling studies and requires further investigation. Especially in light of a recent study published on biorXiv (Kuper et al., 2024) where the authors show functional impact of the plug region in addition to a structure of XPD engaged with a Y-junction DNA containing a crosslink damage. Their experimental structure also differs from what is presented here. However, this study has not been published in a peer-reviewed journal yet.

- The authors suggest that the bubble could be around 23 bp in size which would match the size of the excised fragment. They also state the damage would be located 8 bases away from the 3' incision which is correct for the cited study. There are, however, other studies like Bessho et al. 1997 that performed analyses on a psoralen monoadduct which was incised 5 bases away from the lesion. This ambiguity in exact lesion position should be reflected in the current interpretation.

- Concerning the modelling of the XPA C-terminus: This has been modelled by Kim et al. previously (Nature 2023). There is thus no need to model this again and not to refer to the previous study.

- With respect to the XPC-XPG competition for p62 binding: Previous studies have been performed on the respective yeast proteins that already suggested this. In fact, the publication cited in the current manuscript for the acidic patch "Structural and functional evidence that Rad4 competes with Rad2 for binding to the Tfb1 subunit of TFIIH in NER" addresses this. This should be clearly mentioned and cited accordingly. The authors current interpretation insinuates that this is their genuine finding. If their acidic region is different from the one described for Rad2 it should be clearly stated.

- The competition between XPC and XPG for p62 seems reasonable and explains observed data. However, why do the authors refrain from modelling the XPG p62 interaction and only model a fraction of p62? After all, this interaction is unlikely to just disappear and the authors also do not state anything otherwise. Since all other subunits were completed and modeled one would expect similar scrutiny for the important recruitment factor p62.

Minor points:

- Writing should be checked and corrected

- Figures should appear in the right order and not for example 2f prior to 2c, d, e etc.

Overall, it seems necessary that the authors more clearly convey that they generated a model that is in some parts supported by published structures and that their modelling relies on many assumptions that have not been proven so far. For a publication in Nature Communications it also seems to be essential that the authors back up their findings with biochemical data. The computational analysis would greatly benefit from it, otherwise the study could be interpreted as being too speculative.

At the current stage publication in Nature Communication can not be supported. The major concerns mentioned above have to be addressed but more importantly the entire lack of biochemical experiments does not provide the confidence required to support the suggested models and a publication in a specialized journal for modeling studies may be more suitable.

Reviewer #3 (Remarks to the Author):

The manuscript by Yan et al. presents an impressive study that integrates cryo-EM and XL-MS data with AlphaFold2 to construct a comprehensive model of the NER pre-incision complex (PInC). This model was further analyzed through extensive MD simulations, offering valuable insights into the global dynamics of this super complex. A notable aspect of this study is the successful construction of such a large, dynamic multicomponent complex using computational modeling and simulation, which was unachievable with experimental methods alone. The manuscript also contextualizes the predicted structures within the framework of existing studies. Furthermore, the mapping of disease mutations onto their model provides unique mechanistic insights into the etiology of xeroderma pigmentosum and Cockayne syndrome. Overall, the study is well-conceived and executed, and the manuscript is excellently written. I have a few suggestions for the authors to consider:

1. It would be helpful if the authors could provide some basic statistics about their final model. For instance, what is the percentage of the final structural model that was modeled ab initio? How much of these modeled structural elements is located at the protein interfaces? How much of the model involves significant structural rearrangements during modeling? What are the conformational differences of the individual proteins in PIC as compared to those in other structures?
2. While the authors have thoroughly validated their model by comparing it to existing experimental data, is there a method to quantify the uncertainty of the modeled structure, such as using the pLDDT score in AlphaFold?
3. Given the size of the manually constructed complex, how did the authors determine the duration of the MD simulation needed to capture functionally relevant dynamics? How did the authors distinguish between relaxation dynamics and functionally relevant dynamics? How was the convergence of the sampled dynamics determined?
4. The authors should provide more clarity on how cross-linking mass spectrometry (XL-MS) data were used to guide the modeling protocol in creating a complete structural model of the human

NER pre-incision complex. Could the authors provide more details on how these restraints were applied, how many were applied, and how many were satisfied in their final model?

5. A minor point to note is that GH1 (residues 33-41) is not depicted anywhere in Figure 2.

We thank you and the three reviewers for their careful consideration of the manuscript and their positive feedback.

Point by point response to reviewer comments:

REVIEWER COMMENTS

Reviewer #1 (Remarks to the Author):

Yu et al propose a molecular model of the human nucleotide excision repair complex by synthesising available structural information (chemical crosslinks, experimental structures), addition of AlphaFold2 modelling and molecular dynamics simulations. They use the obtained model to propose a lesion recognition mechanism and interpret distinct classes of human disease mutations in various subunits. The overall theme of the research, the architecture and mechanism of the highly dynamic and complex nucleotide excision machinery is a timely topic of broad and general interest. A validated, high-quality model of the machinery is a valuable resource for the field and beyond. The computational tools are reasonably well described and are state of the art. The paper is generally well written and covers a lot of ground.

The authors derive interesting conclusions from the model with respect to lesion recognition and cleavage as well as disease mutations.

I have two main points regarding the work in this manuscript that need to be addressed before publication.

1) No independent experimental validation. While the molecular model is consistent with available information, it was not validated by a model-derived testable hypothesis. I am aware that this is quite a challenge for such a complex system and perhaps beyond reach at this stage. However, perhaps the authors could at least perform a robust interface analysis that provides orthogonal measures for the quality of experimentally known (old) and modelled (new) interfaces (stereochemistry, packing quality, chemical complementarity etc.

As noted by Reviewer #1, detailed experimental validation of the PInC integrative model and all interfaces therein is beyond the scope of this paper. The value of the model is in the novel mechanistic ideas it provides, which could be tested by other researchers in the NER field. To this end, we have deposited model coordinates in PDB-dev, which will be made available upon publication.

We followed the reviewer's suggestion and examined the quality of the interfaces in the PInC model.

1. The deposited PInC model has undergone real space refinement with the Phenix package followed by local refinement with Coot. We did not observe any obvious issues with stereochemistry, clashes, or significant molecular geometry violations for the interfacial residues (see validation table below and a screenshot from the Coot session; green bars along the protein chains indicate the geometry is OK for the particular residues shown):

Validation statistics after real space refinement	PInC model	[redacted]
MolProbity score	2.42	
MolProbity Clashscore	20.5	
Rotamers outliers (%)	0.45	
C β deviations (%)	0.02	
Ramachandran favored (%)	87.25	
Ramachandran allowed (%)	11.65	
Ramachandran outliers (%)	1.10	

2. We added a new Supplementary Figure 11 to show that the modeled interfaces exhibit good geometric and electrostatic complementarity as well as conservation.

3. We performed interface analysis for the important XPD-XPG and XPF-ERCC1, ERCC1-XPA and XPF-XPD interfaces with the PISA server. Results are summarized below.

Interface	XPG		XPD	
	Number of atoms	323	7.1%	341
Number of residues	83	15.1%	85	11.2%
Interface surface area, Å ²	3078.8			
gain on complex formation, kcal/mol	-23.0			
N _{HB}	21			
N _{SB}	9			

Interface	ERCC1		XP F	
	Number of atoms	314	19.9%	336
Number of residues	80	40.4%	95	41.9%
Interface surface area, Å ²	3066.0			
gain on complex formation, kcal/mol	-41.0			
N _{HB}	22			
N _{SB}	4			

Interface	ERCC1		XPA	
	Number of atoms	182	11.5%	155
Number of residues	48	24.2%	37	13.6%
Interface surface area, Å ²	1526.9			
gain on complex formation, kcal/mol	-17.7			
N _{HB}	14			
N _{SB}	0			

Interface	XPF		XPD	
	Number of atoms	78	4.4%	87
Number of residues	19	8.4%	24	3.2%
Interface surface area, Å ²	757.8			
gain on complex formation, kcal/mol	-2.9			
N _{HB}	6			
N _{SB}	0			

** N_{HB} = Number of hydrogen bonds

2) I think it is ok to perform the *in silico* analysis of the disease mutations on top the (unvalidated) model, however I find the proposed repair mechanism too detailed and speculative at this stage without experimental validation or new experimental data. It would be ok though to add a schematic panel as a simplified conclusion from the model.

We have revised and simplified the schematic in Fig. 8 to place the focus on the main conclusions of paper rather than the overall repair mechanism. Early- and late-stage steps in the NER mechanism have been omitted for clarity.

Other points for revising the manuscript:

1) It would be tremendously helpful to provide in the supplement an overview figure highlighting on their pre-incision complex model, which part is derived from experimental structures, which parts are modelled by which computational models. For instance, a replica of Fig 1 but showing the source of information for each subparts.

Thank you for this suggestion. We have added a new Supplementary Figure 9 – a replica of Fig. 1 identifying the source of experimental information for each part of the model. We have also summarized this information in greater detail in a new Supplementary Table 2.

Table S2 Summary of PInC structural elements and original sources used for hybrid modeling

Protein	Chain	Size (aa)	Modeled Residues	Alternative names	Structures (PDB IDs) used for hybrid modeling ^a
XPB	A	782	34-203,248-720	ERCC3	Modeled from 6RO4
XPD	B	760	1-760	ERCC2	Modeled from 6RO4
p52	C	462	18-458	GTF2H4	Modeled from 6RO4
p44	D	395	11-387	GTF2H2	Modeled from 6RO4
p34	E	308	1-292	GTF2H3	Modeled from 6RO4
p8	F	71	2-67	GTF2H5	Modeled from 6RO4
XPA	G	273	1-273		Residues 104-234 modeled from 6RO4; The XPA N-terminal extension (residues 1-103) and the C-terminal extension (b-domain) (residues 235-273) modeled using AlphaFold2; Residues 22-40 was positioned by the X-ray structure of RPA32C/Smardc11 N-terminus (4MQV). Residues 67-77 modeled from NMR structure (2JNW)

p62	H	548	395-548	GTF2H1	Modeled from 6O9M and cryo-EM density (EMD-4970)
XPG	I	1186	1-296,733-985	ERCC5	DNA-bound XPG modeled based on the human FEN1/DNA X-ray structure (5UM9). The two XPG gateway helices (GH2 residues 82-129) and the capping helix (CH, residues 734-763) were predicted with AlphaFold2 and positioned in the gap between XPD's Arch and Fe-S domains in accordance with the crosslink data. The XPD-anchor domain (residues 157-296) was predicted by AlphaFold2 and positioned based on TFIIH/XPA/DNA cryo-EM density (EMD-4970).
XPF	J	916	680-906	ERCC4	Modeled from cryo-EM structures of XPF/ERCC1 (6SXA and 6SXB). Mg ²⁺ ion coordination was based on the Aeropyrum pernix SNF2 structure (2BGW).
ERCC1	K	297	100-297		Modeled from cryo-EM structures of XPF/ERCC1 (6SXA and 6SXB).
RPA70	L	616	183-616		RPA70AB/ssDNA (residues 183-415) was modeled by the yeast RPA/ssDNA structure (1JMC) and human apo-RPA 70AB (6I52). RPA70C/ssDNA (residues 442-596) was modeled by Ustilago maydis RPA/ssDNA structure (4GOP) and human trimer core structure (1L1O).
RPA14	M	121	3-117		Modeled from human trimer core structure (1L1O).
RPA32	N	270	44-268		Modeled from human trimer core structure (1L1O).

2) The depiction of crosslinks in Fig. 2 is not ideal since one only sees crosslinked residues but not the path of the crosslinks. Perhaps it gets too confusing, but I suggest to add a panel, e.g. in the supplement that shows the paths of the crosslinks on the structures. Furthermore, a bit more detail is needed here, to what extent do crosslinks agree with the structure at a given cutoff for the type of crosslinker used.

We appreciate this suggestion. Upon inspection we found that showing the paths of the crosslinks does not make panel b of Figure 2 too crowded. Therefore, we have added the paths of the crosslinks directly to the new Figure 2. The cutoff length for the crosslinks, in this particular case, was 30 Angstroms. Cross-links were filtered with scores above 6, which is the same threshold used in Kocic, G. et al. Nat Commun 10, 2885 (2019).

3) The color scheme for the disease mutations in the movies is not very clear, the white or light pink spheres are practically invisible and what stand out are only the dark blue or black spheres. Is this intended? If not I suggest a scheme based on variations on hue but keeping saturation similar so that all proposed groups are equally visible?

We have changed the movies to make the mutant positions stand out more. We have changed the color of the pink spheres, switched from ambient to two-point lighting and employed depth cueing to improve clarity.

4) XPD is a 5' -> 3' helicase correct? I am confused by the sentence on p16 "XPD reels in ssDNA in the 3' to 5' direction", please clarify or reformulate.

XPD is a 5' -> 3' helicase and moves on DNA from the 5' toward the 3' end when unimpeded. When the position of XPD is fixed (as in the TFIIH complex) then the ssDNA is moved toward the 5' junction of the NER bubble. We have clarified this in the text of the manuscript: "XPD reels in ssDNA toward the 5' junction".

Reviewer #2 (Remarks to the Author):

Yu et al. present a modeling study on the handover of the XPC/centrin2/Rad23B complex after initial lesion recognition and the handoff to TFIIH as central factor in the pre-incision complex responsible for lesion verification and positioning of the nucleases as well as priming them for

incision. This study is a follow up on their previous study from 2023 and now includes additional proteins and steps in the NER process. The authors utilized a combination of cryo EM and crystal structures, interpreted additional densities in cryo EM maps, and modeled those based on existing additional data. The modeled complexes were then subjected to further computational analyses to display global motions, dynamic communities and towards the impact of disease related mutations. Overall, the study provides interesting perspectives and possibilities on how a pre-incision bubble may be formed, how the size of the bubble is determined, and how the nucleases are positioned for incision. The authors describe potential interactions and possible molecular crosstalk between subunits like XPG and XPD, XPA and XPF/ERCC1, XPG and p62. They also provide a plausible model for RPA protecting the undamaged DNA strand.

This is an interesting study provoking thoughts how core TFIIH may function and could crosstalk to other NER factors. However, there are some serious concerns which prevent a recommendation for publication in Nature Communication:

- Overall, the study is highly speculative. None of the proposed complexes and most importantly protein-protein interactions and protein-nucleic acid interactions were further analyzed or verified by biochemical experiments.

Please see our detailed response to Reviewer #1.

Additionally, we note that our model synthesizes available experimental data and combines it with AlphaFold2 predicted structural modules and interfaces. Therefore, the model reveals emergent properties that could not have been construed by examining the PInC constituent parts in isolation and could not have been derived directly from previous experiments. In fact, as the reviewer notes: “... the model provides interesting perspectives and possibilities on how a pre-incision bubble may be formed, how the size of the bubble is determined, and how the nucleases are positioned for incision.”

Thus, the results from our integrative modelling and dynamics simulations are novel and support a structural basis for XPF and XPG nuclease licensing and coordination for PInC dual incision.

- The authors claim to provide a model leading from initial damage recognition to damage verification. However, they entirely ignore the required dissociation of the CAK complex. This is an absolutely necessary step in the NER process. The detailed description of the individual steps within the manuscript implies that they have been all addressed which is clearly not the case.

We do not make such a claim. In the Introduction section (on page 4 of the manuscript) we clearly state that our findings concern late-stage NER from lesion scanning to dual incision: “..., thus shedding light on the reorganization of the NER protein machinery from the middle through the late stages of the pathway.”

Yet, we reasoned that some readers may benefit from seeing our results discussed in the context of a more complete description of the NER mechanism. Thus, the schematic outline of the mechanism in the original Fig. 8 included several early intermediates based on current mechanistic knowledge in the field. To prevent any misconception regarding the claims of the paper, we changed Fig. 8 and removed all references to initial damage recognition and the early stages of the repair mechanism.

The description of the mechanism under Discussion is focused on late-stage NER and is by no means intended to be comprehensive. Yet, some mention of lesion recognition is warranted in order to provide context to readers who may not be intimately familiar with the NER field. Such context is kept to a minimum in the revised manuscript. The required dissociation of the CAK complex is now noted to be a separate process that needs to be independently considered.

- The additional modelling performed by the authors based on the map of *bro4* is very biased towards XPG replacing *p62* at the anchoring position. The crosslinking data from previous studies support that XPG can bind in that area but there is also data supporting the presence of *p62*. Inspection of the residual density modelled in this study shows no additional density that supports the presence of XPG rather than *p62* thus making their interpretation even less likely. Furthermore, the crosslinking data do not show which kind of particles went into the final map of the cryo EM structure. It is thus unclear if XPG or *p62* are present in the map. Furthermore, in an earlier study by Barnett et al. NAR 2020 (7ad8) *p62* was built into the residual density which should be discussed here. The authors can argue that this is a mutual binding site for the anchor, it is however highly unlikely that this is exclusively XPG or XPG at all, since there is clear evidence for the presence of *p62* and virtually none for the presence of XPG. A detailed map of the crosslinks mentioned with respect to *p62* and XPG in that region of XPD would also be appreciated and seems necessary.

As suggested, we now show a schematic mapping of crosslinks onto the domain structure of XPG, XPD and *p62*. We also provide a table supporting our XPG placement, which includes all XPG-XPD, *p62*-XPD and *p62*-XPG crosslinks with scores above 6 (the filtering threshold used in the original study)

It is true that XL-MS data are subject to conformational and compositional variability and may, in principle, diverge from the dominant cryo-EM structure. However, in this case, there is excellent correspondence between the 6RO4 cryo-EM structure and the XL-MS data set, which was used for validation in Kocic et al. (2019). In that study, >85% of the crosslinks were within the 30-A cutoff.

XPG	XPD	Score
98 (GH2)	184 (Fe-S)	7.46
98 (GH2)	271 (Arch)	13.53
98 (GH2)	344 (Arch)	11.67
107 (GH2)	131 (Fe-S)	7.97
107 (GH2)	344 (Arch)	16.01
123 (GH2)	131 (Fe-S)	12.57
123 (GH2)	184 (Fe-S)	8.51
123 (GH2)	373 (Arch)	6.99
123 (GH2)	344 (Arch)	6.82
129 (GH2)	131 (Fe-S)	9.61
129 (GH2)	271 (Arch)	17.74
129 (GH2)	344 (Arch)	6.13
129 (GH2)	373 (Arch)	6.15
157 (XPG Anchor)	271 (Arch)	9.57
166 (XPG Anchor)	131 (Fe-S)	7.29
175 (XPG Anchor)	131 (Fe-S)	7.08
175 (XPG Anchor)	271 (Arch)	9.97
175 (XPG Anchor)	344 (Arch)	6.13
216 (XPG Anchor)	131 (Fe-S)	8.09
277 (XPG Anchor)	344 (Arch)	13.02
296 (XPG Anchor)	344 (Arch)	10.57
436 (Disordered region)	277 (Arch)	7.69
(Disordered region)	271 (Arch)	10.66
p62	XPD	Score
41 (PH domain)	131 (Fe-S)	6.69
243 (BSD)	131 (Fe-S)	6.47
299 (XPD anchor)	184 (Fe-S)	7.74
308 (XPD anchor)	116 (Fe-S)	9.64
p62	XPG	Score
41 (PH domain)	98 (GH2)	9.64
41 (PH domain)	757 (CH)	7.49

Notably, there are 23 XPD-XPG with scores ranging from 17.74 to 6. By contrast, there were only 4 p62-XPD crosslinks. One connects the p62 PH domain (res 41) and the XPD Fe-S domain (res 131), which is consistent with our placement of the PH domain as shown in Supplementary Fig. 4. The other three crosslinks involve BDS2 and the p62 XPD-anchor regions but have marginal scores. Thus, the preponderance of XL-MS evidence points to XPG being bound to XPD and not p62.

Second, we compared how well the XPG-anchor in our model fits the unmodeled region of the 6RO4 cryo-EM density relative to 1) our AlphaFold2 docked p62-XPD complex; and 2) the p62 region fitted in the NAR 2020 (7ad8) paper. We note that the XPG-anchor (panels a-c) fits the density much better than the other two models. Strikingly, this fit can be achieved without backbone modifications of the AF2 structure. We also note that the p62 AlphaFold2-predicted (panels d-f) and 7AD8 (panels g-i) models cover different regions in the p62 sequence, with the 7AD8 region showing low helical propensity (panel h).

Finally, we calculated EMRinger scores for the respective XPD-p62 and XPG-XPD complexes in the EM density. The EMRinger scores estimate how well the modelled side chains fit the density. For the XPD-XPG complex the EM-ringer score was 1.85, indicating correct placement of the majority of side chains at the XPD-XPG interface. For the two p62 complexes we found values of -1.4 and 1.07, suggesting poor residue fitting.

Figure (for review purposes). Comparison of fit to the EMD-4970 density of a-c) XPD–XPG anchor complex; d-f) p62-XPD complex modeled by AlphaFold2; and g-i) p62-XPD complex from PDB ID: 7AD8. Overall complexes are shown in panels a, d and g. Zoomed-in views of the interfaces with XPD are shown in panels c, f and i.

Once the manuscript is published the complete coordinates will be archived in PDB-dev and our experimentally based and testable placement can then be verified by researchers worldwide.

- The here provided coordinate model displays the XPD plug region, that has been shown to be of functional importance, modeled very closely to the apo form of TFIIH that still contains MAT1. In *bro4* this part of the XPD molecule is disordered and Kokic et al. argue that releasing the plug is activating XPD helicase. This seems to be contradictory to the modelling studies and requires further investigation. Especially in light of a recent study published on *bioRxiv* (Kuper et al., 2024) where the authors show functional impact of the plug region in addition to a structure of XPD engaged with a Y-junction DNA containing a crosslink damage. Their experimental structure also differs from what is presented here. However, this study has not been published in a peer-reviewed journal yet.

Thank you for pointing us to the Kuper et al. *bioRxiv* paper (now published in NSMB). The plug region of XPD is clearly important, but also a highly mobile/variable region. Modeling XPD with AlphaFold2 in isolation shows the plug to be consistent with the conformation found in the preinitiation complex (see panel (a) below). However, the low pLDDT score suggests that the confidence of the fold prediction is low for this region. We also have carried out independent simulations of XPD (not discussed in this manuscript) in nucleotide-bound and apo states and observed that the XPD plug is mobile and also undergoes partial unfolding (see panel (b) below). We now note the importance and the mobility of the XPD plug in the revised manuscript and cite the Kuper et al. paper.

Figure (for review purposes). a) XPD-plug region modeled by AlphaFold2 with pLDDT score color coded onto the structure; b) Two conformations of the XPD-plug taken from MD simulations of XPD. The structures show mobility and partial unfolding.

- The authors suggest that the bubble could be around 23 bp in size which would match the size of the excised fragment. They also state the damage would be located 8 bases away from the 3' incision which is correct for the cited study. There are, however, other studies like Bessho et al. 1997 that performed analyses on a psoralen monoadduct which was incised 5 bases away from the lesion. This ambiguity in exact lesion position should be reflected in the current interpretation.

Thank you for this suggestion. Small variability in the lesion position relative to the 3' junction has been observed biochemically and in single-molecule spectroscopic studies. We now note this point in the revised version and cite the Bessho et al. paper.

- *Concerning the modelling of the XPA C-terminus: This has been modelled by Kim et al. previously (Nature 2023). There is no need to model this again and not to refer to the previous study.*

The XPA C-terminal beta sheet was indeed resolved in the Nature 2023 study, and we provide the corresponding citation (reference #16 Kim, J. et al. Nature 617, 170-175 (2023)). However, the placement of XPA in our model was done prior to the paper becoming available. Therefore, this segment of XPA was modelled from AlphaFold2 and positioned based on the EMD-4970 cryo-EM density as described under Methods.

- *With respect to the XPC-XPG competition for p62 binding: Previous studies have been performed on the respective yeast proteins that already suggested this. In fact, the publication cited in the current manuscript for the acidic patch “Structural and functional evidence that Rad4 competes with Rad2 for binding to the Tfb1 subunit of TFIIH in NER” addresses this. This should be clearly mentioned and cited accordingly. The authors current interpretation insinuates that this is their genuine finding. If their acidic region is different from the one described for Rad2 it should be clearly stated.*

As noted, the XPC-XPG competition for binding to the p62 PH domain is not a new finding. This was clearly stated and cited in the original version: “The fact that XPC and XPG both compete for binding to the 3' DNA duplex and do not coexist in NER complexes [61,62], ... and requires XPC binding to p62's PH domain via a conserved acidic patch [59]”

We have provided the corresponding references:

59. Lafrance-Vanasse, J., et al. Nucleic Acids Res 41, 2736-45 (2013).

61. Wakasugi, M. & Sancar, A. Proc Natl Acad Sci USA 95, 6669-74 (1998).

62. Riedl, T., Hanaoka, F. & Egly, J.M. EMBO J 22, 5293-303 (2003).

The new finding is that the XL-MS data support placement of the p62 PH-domain proximal to the newly modeled XPG-anchor, XPG coiled-coil helices (GH2 and CH) and the Fe-S domain of XPD (Supplementary Fig. 4). The XPG acidic patch (human XPG residues 151-164) is also positioned at the edge of the XPG-anchor domain. Such spatial proximity of the PH domain to the XPG acidic patch could not have been construed from the previous studies. Our acidic region aligns to residues 128-148 in the Rad2 sequence, which is indeed different from the one described in reference #59. Furthermore, as noted in reference #59, the XPG spacer region has multiple acidic segments that compete for p62 binding. This is now clarified in the revised manuscript.

- *The competition between XPC and XPG for p62 seems reasonable and explains observed data. However, why do the authors refrain from modelling the XPG p62 interaction and only model a fraction of p62? After all, this interaction is unlikely to just disappear and the authors also do not state anything otherwise. Since all other subunits were completed and modeled one would expect similar scrutiny for the important recruitment factor p62.*

We modelled only the fraction of p62 (p34-anchor and 3-helix bundle domain; residues 395-547) that could be positioned with confidence into the EMD-4970 cryo-EM density. P62 is highly

flexible, except for structured segments forming interfaces with TFIID. This was precisely the reason why p62 was not modeled into the EM density in the original Kokic et al.(2019) study.

Additionally, we have no experimental or computational evidence that p62 binds XPG after displacement from XPD. We speculate that the released p62 segments (p62 anchor helices and BSD2 domain) may remain flexible near the XPG-anchor and the XPD Fe-S domain, but the residual density in this region is not strong enough for structural interpretation. This point will no doubt be further investigated by ongoing cryo-EM efforts.

Minor points:

- *Writing should be checked and corrected*

Done.

- *Figures should appear in the right order and not for example 2f prior to 2c, d, e etc.*

Done.

Overall, it seems necessary that the authors more clearly convey that they generated a model that is in some parts supported by published structures and that their modelling relies on many assumptions that have not been proven so far. For a publication in Nature Communications it also seems to be essential that the authors back up their findings with biochemical data. The computational analysis would greatly benefit from it, otherwise the study could be interpreted as being too speculative.

We added a new figure and supplementary table showing which parts of the model were based on AlphaFold2 and which were based on experimental structures (with their respective PDB IDs). We believe that the computational modeling, the dynamics simulations as well as the dynamic network analysis are important data in themselves.

At the current stage publication in Nature Communication can not be supported. The major concerns mentioned above have to be addressed but more importantly the entire lack of biochemical experiments does not provide the confidence required to support the suggested models and a publication in a specialized journal for modeling studies may be more suitable.

The reviewer makes a strong philosophical point, and we certainly recognize the value experimental data for unraveling biological mechanisms. However, we do not share the view that new biochemical experiments are a prerequisite for publishing in broad multidisciplinary journals such as Nature Communications.

Consider the following:

1. Even top journals with clear experimental leaning (e.g., Cell) provide avenues to publish papers that yield new conceptual advances by synthesizing and reinterpreting existing experimental data. Many computational biology, bioinformatics and genomics articles fall into this category, specifically when they analyze existing data collected by various genomic consortia.

Specifically, in our study, we have:

1) Interpreted cross-linking data that had received only partial interpretation in the original EM study by Kokic et al. Nat Commun 10, 2885 (2019).

2) Added previously unmodeled regions based on the available cryo-EM density (EMD-4970) and AlphaFold2 calculations. This could not have been accomplished at the time of the original cryo-EM study.

3) Used existing cryo-EM, X-ray and NMR structures of constituent proteins and subassemblies to produce a practically complete integrative model of PInC that enabled quantitative evaluation of its functional dynamics.

Furthermore, consider the following excerpt from www.cell.com/cell/article-types: “Papers that employ computational, theoretical, or analytical approaches to derive novel conceptual models with clearly experimentally testable predictions.” Indeed, the point of integrative modeling is to synergistically combine existing data from multiple experimental sources to yield conceptual advances that could not have been achieved by any single experimental technique. Collectively the integrative model and our computational analyses provide specific and testable hypotheses regarding PInC’s organization, conformational switching and regulation and moreover explain different genetic disease mutational phenotypes.

To highlight the value of the modelling we have added the following text to the Discussion section: “Linking molecular mechanisms to disease phenotypes is a grand challenge for structural biology. This challenge is often unmet as it requires knowledge of dynamic conformations and assemblies that resist purely experimental approaches. Our integrative methods and results provide a framework for meeting this challenge and for designing future experiments to uncover the intricate molecular choreography of global genome NER.”

Reviewer #3 (Remarks to the Author):

The manuscript by Van et al. presents an impressive study that integrates cryo-EM and XL-MS data with AlphaFold2 to construct a comprehensive model of the NER pre-incision complex (PInC). This model was further analyzed through extensive MD simulations, offering valuable insights into the global dynamics of this super complex. A notable aspect of this study is the successful construction of such a large, dynamic multicomponent complex using computational modeling and simulation, which was unachievable with experimental methods alone. The manuscript also contextualizes the predicted structures within the framework of existing studies. Furthermore, the mapping of disease mutations onto their model provides unique mechanistic insights into the etiology of xeroderma pigmentosum and Cockayne syndrome. Overall, the study is well-conceived and executed, and the manuscript is excellently written.

I have a few suggestions for the authors to consider:

1. It would be helpful if the authors could provide some basic statistics about their final model. For instance, what is the percentage of the final structural model that was modeled ab initio? How much of these modeled structural elements is located at the protein interfaces? How much of the model involves significant structural rearrangements during modeling? What are the conformational differences of the individual proteins in PIC as compared to those in other structures?

The new Supplementary Figure 9 and Supplementary Table 2 give an idea as to which regions were modeled from known structures, and which were modeled ab initio. The AF2-modeled

regions (about 400 residues) constitute a relatively small fraction of the PInC complex (4882 residues total). However, new interfaces involving these regions as well as regions modeled from known structures are quite substantial. Please refer to our response to Reviewer #1 regarding assessment of these newly modeled interfaces. There are regions, including the XPG-spacer and XPF/ERCC1, which have undergone significant conformational rearrangements. Yet, most TFIIH subunits were modeled directly from the from 6RO4 cryo-EM structure with only minor modifications as indicated in Supplementary Table 2.

2. While the authors have thoroughly validated their model by comparing it to existing experimental data, is there a method to quantify the uncertainty of the modeled structure, such as using the pLDDT score in AlphaFold?

Thank you for this suggestion. We have added a new Supplementary Figure 10 which shows the regions modelled with AlphaFold2 with their respective pLDDT scores mapped onto the structures.

Supplementary Figure 10. Regions of the PInC assembly modeled with AlphaFold2 with pLDDT scores mapped onto the structures. a) XPG-anchor-XPB complex; b) p62 BSD2 and XPB-anchor in complex with XPB; c) XPA C-terminal and N-terminal regions.

4. Given the size of the manually constructed complex, how did the authors determine the duration of the MD simulation needed to capture functionally relevant dynamics? How did the authors distinguish between relaxation dynamics and functionally relevant dynamics? How was the convergence of the sampled dynamics determined?

To deal with relaxation dynamics, we relied on the change in the RMSD from the initial model (computed over C- α in and P atoms). We monitored RMSD convergence and excluded from analysis the initial trajectory frames up to the convergence point (~60 ns).

Regarding functional dynamics, we were guided by a previous study (Yu, J. et al. Nat Commun 14, 2758 (2023)) in which we had simulated the TFIIH-NER lesion scanning complex and found that upon circularization and closure of TFIIH XPB dynamics was suppressed while XPD showed multiple opening and closing events at the gap between its Arch and Fe-S domains, indicative of its ability to translocate DNA. In the current study, we were once again principally concerned with XPB and XPD translocase dynamics and how these dynamics were influenced by XPG binding. Specifically, we wanted to find out whether the opening/closing dynamics of XPD was still present or not. Therefore, the MD simulations in our current study were carried out on a timescale commensurate with the Yu, J. et al. (2023) study. Of course, one cannot exclude the possibility that much longer simulations may reveal additional functionally relevant motions. However, such motions are beyond the scope of the present manuscript.

5. The authors should provide more clarity on how cross-linking mass spectrometry (XL-MS) data were used to guide the modeling protocol in creating a complete structural model of the human NER pre-incision complex. Could the authors provide more details on how these restraints were applied, how many were applied, and how many were satisfied in their final model?

The XPG-anchor region was created with AF2 and fitted into the EMD-4970 cryo-EM density. The 8 cross-links between the XPG-anchor domain and XPD were only used to validate this positioning. The XPG coiled-coil helices (GH2 and CH) were docked onto XPD using the Expert interface of HADDOCK version 2.4. For docking, we applied a crosslinking cutoff of 30 Å (C α -C α) to the 9 cross-linked pairs connecting the GH2 and CH helices to XPD. Center-of-mass restraints were enabled, and other parameters were set to their default values in HADDOCK. 5 crosslinks were satisfied within the cutoff.

6. A minor point to note is that GH1 (residues 33-41) is not depicted anywhere in Figure 2. Thank you. Fig. 2 was corrected in the revised version.

We appreciate the reviewers' efforts in considering our revised manuscript. Thank you for considering our paper for Nature Communications.

Sincerely,

Ivaylo Ivanov

REVIEWER COMMENTS

Reviewer #1 (Remarks to the Author):

The authors made a reasonable effort to address my points and appropriately revised the manuscript. I wonder why the authors have not included AlphaFold3 in the revised manuscript to more directly predict large subcomplex-DNA architectures (up to 5000 tokens). AlphaFold3 was released with a very capable, fast and accessible webserver for the community. Specifically the DNA interaction prediction capability of AlphaFold3, while probably not flawless, might be an added value and could have provided additional validation. Disregarding the availability of AlphaFold3, I am happy with the revisions.

Reviewer #2 (Remarks to the Author):

please see attached file

Due to the fact that the remarks from the first review were very differently addressed, reviewer 2 directly responds into the rebuttal letter. The remarks are highlighted in green:

Reviewer #2 (Remarks to the Author):

Yu et al. present a modeling study on the handover of the XPC/centrin2/Rad23B complex after initial lesion recognition and the handoff to TFIIH as central factor in the pre-incision complex responsible for lesion verification and positioning of the nucleases as well as priming the pre-incision. This study is a follow up on their previous study from 2023 and now includes additional proteins and steps in the NER process. The authors utilized a combination of cryo EM and crystal structures, interpreted additional densities in cryo EM maps, and modeled those based on existing additional data. The modeled complexes were then subjected to further computational analyses to display global motions, dynamic communities and towards the impact of disease related mutations. Overall, the study provides interesting perspectives and possibilities on how a pre-incision bubble may be formed, how the size of the bubble is determined, and how the nucleases are positioned for incision. The authors describe potential interactions and possible molecular crosstalk between subunits like XPG and XPD, XPA and XPF/ERCC1, XPG and p62. They also provide a plausible model for RPA protecting the undamaged DNA strand.

This is an interesting study provoking thoughts how core TFIIH may function and could crosstalk to other NER factors. However, there are some serious concerns which prevent a recommendation for publication in Nature Communication:

- Overall, the study is highly speculative. None of the proposed complexes and most importantly protein-protein interactions and protein-nucleic acid interactions were further analyzed or verified by biochemical experiments.

Please see our detailed response to Reviewer #1.

Additionally, we note that our model synthesizes available experimental data and combines it with AlphaFold2 predicted structural modules and interfaces. Therefore, the model reveals emergent properties that could not have been construed by examining the PInC constituent parts in isolation and could not have been derived directly from previous experiments. In fact, as the reviewer notes: "... the model provides interesting perspectives and possibilities on how a pre-incision bubble may be formed, how the size of the bubble is determined, and how the nucleases are positioned for incision."

Thus, the results from our integrative modelling and dynamics simulations are novel and support a structural basis for XPF and XPG nuclease licensing and coordination for PInC dual incision.

Providing a hypothesis that can be tested is an important step in the scientific workflow. The authors, however, decided not to take the next step of testing their hypothesis at least to a certain extent. This reviewer agrees with reviewer 1 that although the models are interesting experimental validation is needed for publication in Nat Comm.

- The authors claim to provide a model leading from initial damage recognition to damage verification. However, they entirely ignore the required dissociation of the CAK complex. This is an absolutely necessary step in the NER process. The detailed description of the

individual steps within the manuscript implies that they have been all addressed which is clearly not the case.

We do not make such a claim. In the Introduction section (on page 4 of the manuscript) we clearly state that our findings concern late-stage NER from lesion scanning to dual incision: "..., thus shedding light on the reorganization of the NER protein machinery from the middle through the late stages of the pathway."

Yet, we reasoned that some readers may benefit from seeing our results discussed in the context of a more complete description of the NER mechanism. Thus, the schematic outline of the mechanism in the original Fig. 8 included several early intermediates based on current mechanistic knowledge in the field. To prevent any misconception regarding the claims of the paper, we changed Fig. 8 and removed all references to initial damage recognition and the early stages of the repair mechanism.

The description of the mechanism under Discussion is focused on late-stage NER and is by no means intended to be comprehensive. Yet, some mention of lesion recognition is warranted in order to provide context to readers who may not be intimately familiar with the NER field. Such context is kept to a minimum in the revised manuscript. The required dissociation of the CAK complex is now noted to be a separate process that needs to be independently considered.

Figure 8 is now much clearer. However, in the edited part of the discussion the authors provide details for the actions of XPA like XPB activation and RPA recruitment. In this section the dissociation of CAK needs to be addressed and mentioned, since arrival of XPA/RPA demarks CAK dissociation. Otherwise, their statements could be highly misleading.

- The additional modelling performed by the authors based on the map of 6ro4 is very biased towards XPG replacing p62 at the anchoring position. The crosslinking data from previous studies support that XPG can bind in that area but there is also data supporting the presence of p62. Inspection of the residual density modelled in this study shows no additional density that supports the presence of XPG rather than p62 thus making their interpretation even less likely. Furthermore, the crosslinking data do not show which kind of particles went into the final map of the cryo EM structure. It is thus unclear if XPG or p62 are present in the map. Furthermore, in an earlier study by Barnett et al. NAR 2020 (7ad8) p62 was built into the residual density which should be discussed here. The authors can argue that this is a mutual binding site for the anchor, it is however highly unlikely that this is exclusively XPG or XPG at all, since there is clear evidence for the presence of p62 and virtually none for the presence of XPG. A detailed map of the crosslinks mentioned with respect to p62 and XPG in that region of XPD would also be appreciated and seems necessary.

As suggested, we now show a schematic mapping of crosslinks onto the domain structure of XPG, XPD and p62. We also provide a table supporting our XPG placement, which includes all XPG-XPD, p62-XPD and p62-XPG crosslinks with scores above 6 (the filtering threshold used in the original study)

This figure helps a lot.

It is true that XL-MS data are subject to conformational and compositional variability and may, in principle, diverge from the dominant cryo-EM structure. However, in this case, there is excellent correspondence between the 6RO4 cryo-EM structure and the XL-MS data set, which was used for validation in Kokic et al. (2019). In that study, >85% of the crosslinks were within the 30-Å cutoff.

Notably, there are 23 XPD-XPG with scores ranging from 17.74 to 6. By contrast, there were only 4 p62-XPD crosslinks. One connects the p62 PH domain (res 41) and the XPD Fe-S domain (res 131), which is consistent with our placement of the PH domain as shown in Supplementary Fig. 4. The other three crosslinks involve BDS2 and the p62 XPD-anchor regions but have marginal scores. Thus, we the preponderance of XL-MS evidence points to XPG being bound to XPD and not p62.

Second, we compared how well the XPG-anchor in our model fits the unmodeled region of the 6RO4 cryo-EM density relative to 1) our AlphaFold2 docked p62-XPD complex; and 2) the p62 region fitted in the NAR 2020 (7ad8) paper. We note that the XPG-anchor (panels a-c) fits the density much better than the other two models. Strikingly, this fit can be achieved without backbone modifications of the AF2 structure. We also note that the p62 AlphaFold2-predicted (panels d-f) and 7AD8 (panels g-i) models cover different regions in the p62 sequence, with the 7AD8 region showing low helical propensity (panel h).

Finally, we calculated EMRinger scores for the respective XPD-p62 and XPG-XPD complexes in the EM density. The EMRinger scores estimate how well the modelled side chains fit the density. For the XPD-XPG complex the EM-ringer score was 1.85, indicating correct placement of the majority of side chains at the XPD-XPG interface. For the two p62 complexes we found values of -1.4 and 1.07, suggesting poor residue fitting.

Figure (for review purposes). Comparison of fit to the EMD-4970 density of a-c) XPD–XPG anchor complex; d-f) p62-XPD complex modeled by AlphaFold2; and g-i) p62-XPD complex from PDB ID: 7AD8. Overall complexes are shown in panels a, d and g. Zoomed-in views of the interfaces with XPD are shown in panels c, f and i.

Once the manuscript is published the complete coordinates will be archived in PDB-dev and our experimentally based and testable placement can then be verified by researchers worldwide.

The analysis of the authors is valid. However, since the authors themselves acknowledge that the mass spec data and the final cryo EM map do not need to overlap, they should acknowledge for the possibility that this could also be p62. Otherwise, they themselves provide an excellent reason why experimental validation is absolutely necessary via functional mutagenesis on the respective interface.

- The here provided coordinate model displays the XPD plug region, that has been shown to be of functional importance, modeled very closely to the apo form of TFIIH that still contains MAT1. In *Bro4* this part of the XPD molecule is disordered and Kocic et al. argue that releasing the plug is activating XPD helicase. This seems to be contradictory to the modelling studies and requires further investigation. Especially in light of a recent study published on *bioRxiv* (Kuper et al., 2024) where the authors show functional impact of the plug region in addition to a structure of XPD engaged with a Y-junction DNA containing a crosslink damage. Their experimental structure also differs from what is presented here. However, this study has not been published in a peer-reviewed journal yet.

Thank you for pointing us to the *Kuper et al. bioRxiv* paper (now published in NSMB). The plug region of XPD is clearly important, but also a highly mobile/variable region. Modeling XPD with AlphaFold2 in isolation shows the plug to be consistent with the conformation found in the preinitiation complex (see panel (a) below). However, the low pLDDT score suggests that the confidence of the fold prediction is low for this region. We also have carried out independent simulations of XPD (not discussed in this manuscript) in nucleotide-bound and apo states and observed that the XPD plug is mobile and also undergoes partial unfolding (see panel (b) below). We now note the importance and the mobility of the XPD plug in the revised manuscript and cite the *Kuper et al. paper*.

Figure (for review purposes). a) XPD-plug region modeled by AlphaFold2 with pLDDT score color coded onto the structure; b) Two conformations of the XPD-plug taken from MD simulations of XPD. The structures show mobility and partial unfolding.

Although acknowledging plug mobility is a step in the right direction, this rather crucial

feature of XPD helicase should obtain more attention and the modeling studies should be disclosed and included.

- The authors suggest that the bubble could be around 23 bp in size which would match the size of the excised fragment. They also state the damage would be located 8 bases away from the 3' incision which is correct for the cited study. There are, however, other studies like Bessho et al. 1997 that performed analyses on a psoralen monoadduct which was incised 5 bases away from the lesion. This ambiguity in exact lesion position should be reflected in the current interpretation.

Thank you for this suggestion. Small variability in the lesion position relative to the 3' junction has been observed biochemically and in single-molecule spectroscopic studies. We now note this point in the revised version and cite the Bessho et al. paper.

- Concerning the modelling of the XPA C-terminus: This has been modelled by Kim et al. previously (Nature 2023). There is this no need to model this again and not to refer to the previous study.

The XPA C-terminal beta sheet was indeed resolved in the Nature 2023 study, and we provide the corresponding citation (reference #16 Kim, J. et al. Nature 617, 170-175 (2023)). However, the placement of XPA in our model was done prior to the paper becoming available. Therefore, this segment of XPA was modelled from AlphaFold2 and positioned based on the EMD-4970 cryo- EM density as described under Methods.

The authors do cite reference 16, however, not in the context of modeling of the XPA C-terminus. The result section is as follows:

XPA's C-terminal end contains an extended helix that acts as a clamp on dsDNA and prevents the dissociation of the upstream duplex from XPB. This helical clamp concludes with an antiparallel beta-sheet, which we modelled with AlphaFold2 (Fig. 4a and 4b). The interaction has been previously identified to anchor the XPA C-terminus to p52 and p8 but without structural detail⁴⁶.

Clearly there is no mention of reference 16 here which should be provided for clarity. The study is from 2023 and obviously has been available since it was cited. Whether modeling was based on a different structure is irrelevant for acknowledging the study in this context.

- With respect to the XPC-XPG competition for p62 binding: Previous studies have been performed on the respective yeast proteins that already suggested this. In fact, the publication cited in the current manuscript for the acidic patch "Structural and functional evidence that Rad4 competes with Rad2 for binding to the Tfb1 subunit of TFIIH in NER" addresses this. This should be clearly mentioned and cited accordingly. The authors current interpretation insinuates that this is their genuine finding. If their acidic region is different from the one described for Rad2 it should be clearly stated.

As noted, the XPC-XPG competition for binding to the p62 PH domain is not a new finding. This was clearly stated and cited in the original version: "The fact that XPC and XPG both compete for binding to the 3' DNA duplex and do not coexist in NER complexes [61,62],

... and requires XPC binding to p62's PH domain via a conserved acidic patch [59]'

We have provided the corresponding references:

59. Lafrance-Vanasse, J., et al. Nucleic Acids Res 41, 2736-45 (2013).

61. Wakasugi, M. & Sancar, A. Proc Natl Acad Sci USA 95, 6669-74 (1998).

62. Riedl, T., Hanaoka, F. & Egly, J.M. EMBO J 22, 5293-303 (2003).

The new finding is that the XL-MS data support placement of the p62 PH-domain proximal to the newly modeled XPG-anchor, XPG coiled-coil helices (GH2 and CH) and the Fe-S domain of XPD (Supplementary Fig. 4). The XPG acidic patch (human XPG residues 151-164) is also positioned at the edge of the XPG-anchor domain. Such spatial proximity of the PH domain to the XPG acidic patch could not have been construed from the previous studies. Our acidic region aligns to residues 128-148 in the Rad2 sequence, which is indeed different from the one described in reference #59. Furthermore, as noted in reference #59, the XPG spacer region has multiple acidic segments that compete for p62 binding. This is now clarified in the revised manuscript.

- The competition between XPC and XPG for p62 seems reasonable and explains observed data. However, why do the authors refrain from modelling the XPG p62 interaction and only model a fraction of p62? After all, this interaction is unlikely to just disappear and the authors also do not state anything otherwise. Since all other subunits were completed and modeled one would expect similar scrutiny for the important recruitment factor p62.

We modelled only the fraction of p62 (p34-anchor and 3-helix bundle domain; residues 395-547) that could be positioned with confidence into the EMD-4970 cryo-EM density. P62 is highly flexible, except for structured segments forming interfaces with TFIIF. This was precisely the reason why p62 was not modeled into the EM density in the original Kocic et al.(2019) study.

This might be true but Alphafold provides complete models for p62 that could have been incorporated in this study.

Additionally, we have no experimental or computational evidence that p62 binds XPG after displacement from XPD. We speculate that the released p62 segments (p62 anchor helices and BSD2 domain) may remain flexible near the XPG-anchor and the XPD Fe-S domain, but the residual density in this region is not strong enough for structural interpretation. This point will no doubt be further investigated by ongoing cryo-EM efforts.

If the authors attempted to model this and failed to receive sensible data this should be mentioned accordingly.

Minor points:

- Writing should be checked and corrected

Done.

- Figures should appear in the right order and not for example 2f prior to 2c, d, e etc.

Done.

Overall, it seems necessary that the authors more clearly convey that they generated a model that is in some parts supported by published structures and that their modelling relies on many assumptions that have not been proven so far. For a publication in Nature Communications it also seems to be essential that the authors back up their findings with biochemical data. The computational analysis would greatly benefit from it, otherwise the study could be interpreted as being too speculative.

We added a new figure and supplementary table showing which parts of the model were based on AlphaFold2 and which were based on experimental structures (with their respective PDB IDs). We believe that the computational modeling, the dynamics simulations as well as the dynamic network analysis are important data in themselves.

At the current stage publication in Nature Communication can not be supported. The major concerns mentioned above have to be addressed but more importantly the entire lack of biochemical experiments does not provide the confidence required to support the suggested models and a publication in a specialized journal for modeling studies may be more suitable.

The reviewer makes a strong philosophical point, and we certainly recognize the value experimental data for unraveling biological mechanisms. However, we do not share the view that new biochemical experiments are a prerequisite for publishing in broad multidisciplinary journals such as Nature Communications.

The reviewer does not consider this point philosophical. This is the principle of hypothesis driven science.

Consider the following:

1. Even top journals with clear experimental leaning (e.g., Cell) provide avenues to publish papers that yield new conceptual advances by synthesizing and reinterpreting existing experimental data. Many computational biology, bioinformatics and genomics articles fall into this category, specifically when they analyze existing data collected by various genomic consortia.

Specifically, in our study, we have:

1) Interpreted cross-linking data that had received only partial interpretation in the original EM study by Kokic et al. Nat Commun 10, 2885 (2019).

2) Added previously unmodeled regions based on the available cryo-EM density (EMD-4970) and AlphaFold2 calculations. This could not have been accomplished at the time of the original cryo-EM study.

3) Used existing cryo-EM, X-ray and NMR structures of constituent proteins and subassemblies to produce a practically complete integrative model of PINC that enabled quantitative evaluation of its functional dynamics.

Furthermore, consider the following excerpt from www.cell.com/cell/article-types: “Papers that employ computational, theoretical, or analytical approaches to derive novel conceptual models with clearly experimentally testable predictions.” Indeed, the point of integrative modeling is to synergistically combine existing data from multiple experimental sources to yield conceptual advances that could not have been achieved by any single experimental technique. Collectively the integrative model and our

computational analyses provide specific and testable hypotheses regarding PInC's organization, conformational switching and regulation and moreover explain different genetic disease mutational phenotypes.

To highlight the value of the modelling we have added the following text to the Discussion section: "Linking molecular mechanisms to disease phenotypes is a grand challenge for structural biology. This challenge is often unmet as it requires knowledge of dynamic conformations and assemblies that resist purely experimental approaches. Our integrative methods and results provide a framework for meeting this challenge and for designing future experiments to uncover the intricate molecular choreography of global genome NER."

This reviewer does appreciate the effort and value of the current study. However, without at least some biochemical validation of major aspects presented here it may not warrant publication in Nat Comm.

Reviewer #3 (Remarks to the Author):

All my concerns have been addressed in the revisions, and I recommend publishing it in its current form.

Associate Editor
Nature Communications

Dear Editor,

We thank you and the three reviewers for their careful consideration of the manuscript and their constructive feedback. Our responses and changes in the text of the manuscript are highlighted in blue font.

Point by point response to reviewer comments:

Reviewer #1 (Remarks to the Author):

The authors made a reasonable effort to address my points and appropriately revised the manuscript. I wonder why the authors have not included AlphaFold3 in the revised manuscript to more directly predict large subcomplex-DNA architectures (up to 5000 tokens). AlphaFold3 was released with a very capable, fast and accessible webserver for the community. Specifically, the DNA interaction prediction capability of AlphaFold3, while probably not flawless, might be an added value and could have provided additional validation. Disregarding the availability of AlphaFold3, I am happy with the revisions.

We thank the reviewer for this suggestion. As noted, AlphaFold3 is now available and features protein–DNA complex prediction. However, the software release was very recent (May 8th 2024) and no stand-alone code was made available. Instead, access to AlphaFold3 is through a website interface, which has limited options for customization. Notably, it is not possible to turn off the use of templates with the current web interface, and therefore one cannot run fully ab initio predictions. Doubtless, these issues will be resolved in the coming months, but for now we prefer to use AlphaFold2.

Reviewer #2 (Remarks to the Author):

Due to the fact that the remarks from the first review were very differently addressed, reviewer 2 directly responds into the rebuttal letter. The remarks are highlighted in green:

1. Providing a hypothesis that can be tested is an important step in the scientific workflow. The authors, however, decided not to take the next step of testing their hypothesis at least to a certain extent. This reviewer agrees with reviewer 1 that although the models are interesting experimental validation is needed for publication in Nat Comm.

We understand the reviewer's point. However, at this stage experimental validation is beyond the scope of the manuscript.

2. Figure 8 is now much clearer. However, in the edited part of the discussion the authors provide details for the actions of XPA like XPB activation and RPA recruitment. In this section the dissociation of CAK needs to be addressed and mentioned, since arrival of XPA/RPA demarks CAK dissociation. Otherwise, their statements could be highly misleading.

Thank you. We have now added the following text to the Discussion section (page 16), which clarifies the role of CAK: “TFIIH's CAK module presents an obstacle to further NER progression and is removed upon subsequent XPA recruitment to the 5' end of the NER bubble. This conformational switch involves displacement of CAK's MAT1 subunit by the N-terminus of XPA. It has been previously proposed that

MAT1 could serve as XPB-XPD spacer and its removal could allow XPD to approach DNA^{15,37}. Additionally, XPA stimulates XPB unwinding and orchestrates initial recruitment of RPA.”

CAK module dissociation is also mentioned in the Introduction (page 3): “seven forming core TFIIH (XPD, XPB, p44, p34, p8, p62, and p52) and three comprising the CAK complex (MAT1, Cdk7 and Cyclin H)³⁰. While CAK is key for TFIIH’s function in transcription, its dissociation from core TFIIH is required for functional NER.”

We hope these statements clarify the point.

3. This figure helps a lot.

Thank you.

4. The analysis of the authors is valid. However, since the authors themselves acknowledge that the mass spec data and the final cryo EM map do not need to overlap, they should acknowledge for the possibility that this could also be p62. Otherwise, they themselves provide an excellent reason why experimental validation is absolutely necessary via functional mutagenesis on the respective interface.

To acknowledge this point we have added the following text to the Results section (page 8) of the manuscript: “The overlay shows that the XPG-anchor domain and the p62 XPD-anchor compete for the same binding site on the XPD surface, consistent with a recent study which modeled p62-XPD interactions.⁶² The cryo-EM density unambiguously shows that the XPG-anchor domain is bound to XPD. Yet, the XL-MS data, in addition to 23 above-threshold XPD–XPG crosslinks, also features 4 p62–XPD crosslinks. This may indicate that 1) XPG and p62 compete for the same XPD binding site; or 2) p62’s anchor helices and BSD2 domain remain flexible near XPD’s Fe-S domain after displacement. Since XL-MS is subject to sample conformational and compositional variability, the presence of minor p62-bound species different from the dominant cryo-EM structure cannot be excluded – a point that remains to be addressed by future studies.”

5. Although acknowledging plug mobility is a step in the right direction, this rather crucial feature of XPD helicase should obtain more attention and the modeling studies should be disclosed and included.

To acknowledge the mobility of the XPD plug region we have added the following text to the results section (page 13): “We also note the plug region of XPD – a functionally important, but also a highly mobile and conformationally variable region of PInC⁷⁰. Modeling XPD with AlphaFold2 shows the plug to be consistent with the conformation found in the preinitiation complex (**Supplementary Fig. 8a**). However, the low pLDDT values suggest low fold prediction confidence for this region. Independent simulations of XPD in isolation also show that the plug is mobile and undergoes partial unfolding (**Supplementary Fig. 8b** and **8c**).”

The above-mentioned XPD simulations are now disclosed in new Supplementary Fig. 8.

6. The authors do cite reference 16, however, not in the context of modeling of the XPA C-terminus. The result section is as follows:

XPA’s C-terminal end contains an extended helix that acts as a clamp on dsDNA and prevents the dissociation of the upstream duplex from XPB. This helical clamp concludes with an antiparallel beta-

*sheet, which we modelled with AlphaFold2 (Fig. 4a and 4b). The interaction has been previously identified to anchor the XPA C-terminus to p52 and p8 but without structural detail*⁴⁶.

Clearly there is no mention of reference 16 here which should be provided for clarity. The study is from 2023 and obviously has been available since it was cited. Whether modeling was based on a different structure is irrelevant for acknowledging the study in this context.

Thank you for pointing out this omission. Reference 16 is now cited in text in the position specified by the reviewer: “This helical clamp concludes with an antiparallel β -sheet^[16], which we modelled with AlphaFold2 (Fig. 4a and 4b).”

7. This might be true but Alphafold provides complete models for p62 that could have been incorporated in this study. If the authors attempted to model this and failed to receive sensible data this should be mentioned accordingly.

The p62 protein chain is extended, flexible and splayed on the surface of TFIIH (as observed in the transcription pre-initiation complex). Therefore, the overall p62 conformation is dictated by TFIIH interactions. Modeling complete p62 or p62–XPD complex with AlphaFold2 is possible but unlikely to yield useful structural results and we did not attempt this.

Modeling full-length p62 could be useful in the context of a complete XPG-p62-TFIIH complex. At minimum, such model will have to include all TFIIH subunits that have known p62 interactions: XPD (760 residues), p44 (395 residues), p34 (308 residues) and p52 (462 residues). P62 itself has 548 residues, and XPG has >1000 residues. Unfortunately, modeling all these protein chains is well outside the practical limit of AlphaFold2-multimer (up to ~2500 residues) imposed by GPU memory limitations.

8. The reviewer does not consider this point philosophical. This is the principle of hypothesis driven science. This reviewer does appreciate the effort and value of the current study. However, without at least some biochemical validation of major aspects presented here it may not warrant publication in Nat Comm.

We understand the reviewer’s position. As noted earlier, experimental validation is beyond the scope of the manuscript.

Reviewer #3 (Remarks to the Author):

All my concerns have been addressed in the revisions, and I recommend publishing it in its current form.

Thank you.

We appreciate the reviewers’ efforts in considering our revised manuscript. Thank you for considering our paper for Nature Communications.

Sincerely,

Ivaylo Ivanov